# Conformal Calibration Transfer

Achref Doula [1] [2]

## Abstract

Conformal prediction converts point predictions into set-valued predictions with coverage guarantees under exchangeability between calibration and deployment data. We study *conformal calibration transfer*, where this requirement fails because labeled calibration is available only in a source space, while prediction sets are needed in a target space linked to the source through *unlabeled paired* observations (e.g., paired modalities or sensor changes). We propose Transported Conformal Calibration (TCC): we transport labeled source calibration into the target space using the paired data, and then correct residual post-transport mismatch using only unlabeled target inputs. We instantiate this correction with two complementary methods: **TCC-KS**, which uses a label-free uncertainty surrogate to detect mismatch and adjust calibration conservatively, and **weighted-TCC**, which reweights transported calibration toward the target domain for improved efficiency when weights are stable. We provide finite-sample target-domain coverage guarantees that adapt to an observable measure of mismatch. Across CIFAR-100-C, Tiny-ImageNet-C, and SEN12MS, we show reliable target-domain coverage transfer without labeled target calibration data, with label-free diagnostics that predict when correction is needed.

## 1. Introduction

Conformal prediction (CP) converts a point predictor into a set-valued predictor with finite-sample, distribution-free marginal coverage (Vovk et al., 2005; Shafer & Vovk, 2008; Lei et al., 2018; Angelopoulos & Bates, 2023). In the standard split-conformal construction, this guarantee is obtained by calibrating a nonconformity score on labeled data that

[1]Technical University of Darmstadt, Darmstadt, Germany [2]Tongji University, Shanghai, China. Correspondence to: Achref Doula <achref.doula@gmail.com>.

*Proceedings of the 43rd International Conference on Machine Learning*, Seoul, South Korea. PMLR 306, 2026. Copyright 2026 by the author(s).

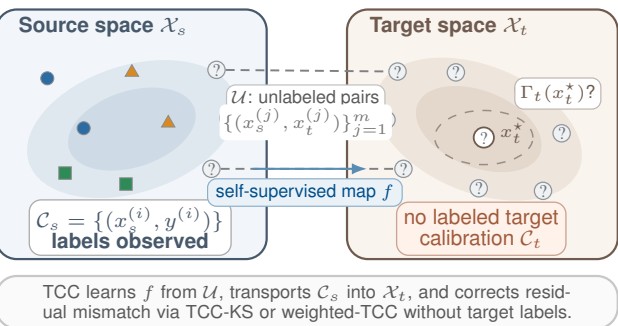

**Problem:** calibration labels are available in $\mathcal{X}_s$, but valid prediction sets are needed in $\mathcal{X}_t$.

Source space $\mathcal{X}_s$ — Target space $\mathcal{X}_t$

$\mathcal{U}$: unlabeled pairs $\{(x_s^{(j)}, x_t^{(j)})\}_{j=1}^m$

$\Gamma_t(x_t^\star)$?

$x_t^\star$

self-supervised map $f$

$\mathcal{C}_s = \{(x_s^{(i)}, y^{(i)})\}$ **labels observed**

no labeled target calibration $\mathcal{C}_t$

TCC learns $f$ from $\mathcal{U}$, transports $\mathcal{C}_s$ into $\mathcal{X}_t$, and corrects residual mismatch via TCC-KS or weighted-TCC without target labels.

*Figure 1.* Conformal calibration transfer. Labeled calibration data are available only in the source space $\mathcal{X}_s$, while prediction sets with target-domain coverage are required in $\mathcal{X}_t$. The two spaces are linked by unlabeled paired observations, but no labeled target calibration set is available.

are exchangeable with the test inputs. In practice, this often means maintaining labeled calibration data for each deployment distribution, even when the task and model are fixed.

In many real deployments, labeled calibration data exist only in a *source* input space $\mathcal{X}_s$, while predictions are required in a *target* space $\mathcal{X}_t$. For instance, a system calibrated on a legacy sensor must be deployed on upgraded hardware, or a system operating on one imaging modality must operate on another (Yi et al., 2021; Lazaridou et al., 2021). When the cost or delay of labeling in the new target space is prohibitive, due to expert annotation requirements, regulatory constraints, or rapid deployment needs, standard conformal calibration is infeasible. However, during sensor transitions or modality changes, collecting *unlabeled paired observations* $(x_s, x_t)$ is often straightforward: co-registered measurements, synchronized captures, or paired preprocessing pipelines naturally yield such data, where each pair corresponds to the same underlying instance and thus shares the same unobserved label.

We formalize this setting as *conformal calibration transfer*. As depicted in Figure 1, we assume access to (i) labeled source calibration $\mathcal{C}_s = \{(x_s^{(i)}, y^{(i)})\}_{i=1}^n$, (ii) a fixed target predictor $g_t : \mathcal{X}_t \to \mathcal{Y}$, and (iii) unlabeled paired data $\mathcal{U} = \{(x_s^{(j)}, x_t^{(j)})\}_{j=1}^m$ linking the two spaces, but crucially *no labeled target calibration data*. The goal is to construct

prediction sets $\Gamma_t : \mathcal{X}_t \to 2^{\mathcal{Y}}$ achieving target-domain marginal coverage $1 - \alpha$ under the distribution of $(X_t, Y)$. Standard split conformal calibration requires labeled target data and is therefore inapplicable. A second natural idea is importance-weighted conformal prediction under covariate shift (Tibshirani et al., 2019), but these methods assume source and target lie in a *common* feature space, which is violated when $\mathcal{X}_s \neq \mathcal{X}_t$. The paired data are therefore the key resource: any successful transfer must exploit the structure that $(x_s, x_t)$ correspond to the same instance.

The unlabeled pairs $\mathcal{U}$ enable learning a transport map $f : \mathcal{X}_s \to \mathcal{X}_t$ that translates source inputs into target counterparts. Given such a map, we can transport the labeled source calibration to form pseudo-target examples $\{(f(x_s^{(i)}), y^{(i)})\}_{i=1}^n$ and apply split conformal calibration under $g_t$. However, transport is generally imperfect: even when $f$ produces plausible target inputs, the transported distribution $\tilde{X}_t = f(X_s)$ may differ systematically from the true target distribution $X_t$ (Hoffman et al., 2018; Wu et al., 2024), shifting the nonconformity score distribution and potentially causing under-coverage. The central challenge is that this residual mismatch must be detected and corrected *without* target labels: we cannot directly compare score distributions because $Y$ is unobserved in the target domain. This limitation is fundamental: we show (Theorem 3.1) that without additional structural assumptions, any procedure guaranteeing distribution-free target coverage using only unlabeled target inputs must be near-vacuous in the worst case. Thus, nontrivial calibration transfer should leverage pairing to produce labeled scores in $\mathcal{X}_t$, and use label-free signals to certify and correct the remaining mismatch.

We propose **Transported Conformal Calibration (TCC)**, a framework that transports source calibration to the target space and then corrects residual mismatch using only unlabeled target inputs. We propose two correction mechanisms. Our first mechanism, **TCC-KS**, provides a certificate-driven adjustment. Since we do not have access to labeled data to compute scores, TCC-KS introduces a *label-free uncertainty surrogate* $T : \mathcal{X}_t \to \mathbb{R}$ computed from the target predictor's outputs (e.g., $T(x) = 1 - \max_y \hat{p}_t(y \,|\, x)$) and measures a one-sided Kolmogorov–Smirnov discrepancy between $T(\tilde{X}_t)$ and $T(X_t)$ on unlabeled samples. A finite-sample high-probability bound yields an observable inflation term $\delta^+$, which we convert into an adjusted level $\alpha^\star = \max(0, \alpha - \delta^+)$: when mismatch is small, $\alpha^\star \approx \alpha$ and the procedure is tight, while large mismatch triggers intentional conservatism. Under an *approximate surrogate control* assumption relating surrogate mismatch to score mismatch, we prove high-probability target coverage $\geq 1 - (\alpha + \varepsilon)$, with slack $\varepsilon$ vanishing when the surrogate tracks true-label difficulty. Our second mechanism, **weighted-TCC**, aims at efficiency by adapting importance-weighted CP to the transfer setting. Unlike standard weighted CP, which reweights

calibration scores from a source distribution in the *same* space, weighted-TCC reweights *after transport*: it treats $\tilde{X}_t = f(X_s)$ as the labeled calibration distribution *in $\mathcal{X}_t$* and estimates density ratios between $\tilde{X}_t$ and $X_t$ from unlabeled samples. This targets the residual shift within the target space and can be efficient when the implied weights are stable. Both TCC-KS and weighted-TCC come with label-free diagnostics, the KS inflation bound and an effective-sample-size stability measure, that can be computed before deployment to guide which correction is trustworthy.

Empirically, on datasets with severe corruptions, namely CIFAR-100-C, Tiny-ImageNet-C (Hendrycks & Dietterich, 2019), and different input domains like SEN12MS (SAR→RGB) (Schmitt et al., 2019), we find that TCC-KS reliably protects coverage when residual mismatch is larger, and weighted-TCC is competitive when residual-shift weights remain stable. We further show that incorporating predictive alignment during transport learning (via a label-free KL term matching $g_t$ outputs) reduces measured mismatch and stabilizes weights, benefiting both correction mechanisms.

## 2. Related Work

**Conformal Prediction and Set-valued Inference.** CP provides distribution-free, finite-sample marginal coverage under exchangeability (Vovk et al., 2005; Shafer & Vovk, 2008; Lei et al., 2018; Angelopoulos & Bates, 2023). In classification, it yields set-valued predictors and connects to controlled-error rules such as least ambiguous classification (LAC) (Sadinle et al., 2019), with practical adoption in deep vision models (Angelopoulos et al., 2021). Distribution-free predictive inference has also been extended beyond standard regression/classification to broader tasks and pipelines (Angelopoulos et al., 2022; Doula et al., 2025; Barber et al., 2021; Romano et al., 2019). Complementary work on calibration emphasizes reliable predictive probabilities under distributional change (Guo et al., 2017; Kuleshov et al., 2018), motivating our focus on preserving coverage under sensor, modality, or processing-pipeline shifts.

**CP under Distribution Shift.** A substantial literature studies CP when calibration and test are not exchangeable. Under covariate shift, importance-weighted conformal methods use density ratios to recover target coverage when the ratio can be estimated from unlabeled target data (Tibshirani et al., 2019; Shimodaira, 2000; Sugiyama et al., 2007). Other settings include feedback covariate shift (Fannjiang et al., 2022) and online or adaptive conformal procedures that update thresholds as distributions drift (Gibbs & Candes, 2021; Gibbs & Candès, 2024). Complementary robust or shift-aware approaches trade efficiency for protection against misspecification or unknown shift (Cauchois et al., 2024; Jeary et al., 2026), and more general risk-control

formulations provide another perspective on controlling error criteria beyond marginal coverage (Angelopoulos et al., 2024). These lines typically assume a single input space for calibration and deployment and therefore do not directly address our paired *cross-space* setting, where calibration labels are available in $\mathcal{X}_s$ but coverage is required in $\mathcal{X}_t$.

**Domain Transport.** Learning mappings between domains is central to domain adaptation. With paired supervision, conditional image translation such as pix2pix learns a map that preserves semantics across appearance changes (Isola et al., 2017). Related unpaired approaches such as Cycle-GAN remove the pairing requirement (Zhu et al., 2017). Domain adaptation more broadly seeks to reduce source–target discrepancy via representation learning or adversarial objectives (Ganin et al., 2016; Ben-David et al., 2010; Kreutz et al., 2024), and transport-based perspectives connect to optimal-transport style alignment criteria (Courty et al., 2017). Source-free domain adaptation adapts predictors using only unlabeled target data (Liang et al., 2020; Yang et al., 2022; Zhang et al., 2022). Across these directions, the goal is typically predictive accuracy or feature alignment rather than distribution-free coverage guarantees for prediction sets. In our framework, the transport map $f$ is used specifically to *move labeled calibration information* into the target input space, after which we quantify and correct residual mismatch to maintain conformal validity.

## 3. Transported Conformal Calibration (TCC)

We study *conformal calibration transfer*, a regime in which labeled calibration data exist only in a source space $\mathcal{X}_s$, while predictions (and conformal coverage) are required in a target space $\mathcal{X}_t$ linked to $\mathcal{X}_s$ through unlabeled paired observations. This section presents (i) a transported-calibration step that leverages these pairs to map labeled source inputs into $\mathcal{X}_t$, and (ii) two complementary, label-free mechanisms to correct residual mismatch between transported and real target inputs: **TCC-KS** and **weighted-TCC**.

### 3.1. Problem Formulation

Let $(X_s, X_t, Y) \sim P$ on $\mathcal{X}_s \times \mathcal{X}_t \times \mathcal{Y}$, where $X_s$ and $X_t$ are paired views and $Y \in \mathcal{Y}$ is the class label. We observe:

- a labeled source calibration set
$$\mathcal{C}_s = \{(x_s^{(i)}, y^{(i)})\}_{i=1}^n, \qquad (x_s^{(i)}, y^{(i)}) \overset{\text{i.i.d}}{\sim} P_{X_s Y},$$

- an unlabeled paired corpus
$$\mathcal{U} = \{(x_s^{(j)}, x_t^{(j)})\}_{j=1}^m, \qquad (x_s^{(j)}, x_t^{(j)}) \overset{\text{i.i.d}}{\sim} P_{X_s X_t}.$$

We consider a fixed target predictor $g_t : \mathcal{X}_t \to \mathcal{Y}$ equipped with a measurable nonconformity score $S_t : \mathcal{X}_t \times \mathcal{Y} \to \mathbb{R}$

(e.g., $S_t(x, y) = 1 - \hat{p}_t(y \mid x)$). For a nominal miscoverage level $\alpha \in (0, 1)$, our goal is to construct prediction sets $\Gamma_t : \mathcal{X}_t \to 2^{\mathcal{Y}}$ with target-domain marginal coverage $1 - \alpha$:

$$\mathbb{P}\{Y \notin \Gamma_t(X_t)\} \leq \alpha, \tag{1}$$

without any labeled target calibration examples.

### 3.2. Transported Split-Conformal Calibration

We first use the paired corpus $\mathcal{U}$ to learn a mapping

$$f : \mathcal{X}_s \to \mathcal{X}_t. \tag{2}$$

We then transport the labeled source calibration inputs and keep labels,

$$\widetilde{\mathcal{C}}_t = \{(\tilde{x}_t^{(i)}, y^{(i)})\}_{i=1}^n, \qquad \tilde{x}_t^{(i)} = f(x_s^{(i)}), \tag{3}$$

compute transported scores $S_t(\tilde{x}_t^{(i)}, y^{(i)})$, and form the empirical $(1 - \alpha)$ quantile

$$\widehat{q}_{1-\alpha} \text{ of } \left\{ S_t(\tilde{x}_t^{(i)}, y^{(i)}) \right\}_{i=1}^n. \tag{4}$$

This yields the familiar split-conformal sets

$$\Gamma_t^{\text{uncorr}}(x) = \{y \in \mathcal{Y} : S_t(x, y) \leq \widehat{q}_{1-\alpha}\}. \tag{5}$$

Transport is generally imperfect: the transported input distribution $\tilde{X}_t = f(X_s)$ may differ from the true target input distribution $X_t$, which can shift the *true-label* score distribution and cause under-coverage. The rest of this section develops two label-free corrections for this residual mismatch using only unlabeled target inputs.

### 3.3. TCC-KS: KS-Corrected Transported Calibration

TCC-KS adjusts the effective calibration level using a label-free discrepancy computed on a surrogate statistic

$$T : \mathcal{X}_t \to \mathbb{R}, \tag{6}$$

derived from $g_t$, such as least-confidence $T(x) = 1 - \max_{y \in \mathcal{Y}} \hat{p}_t(y \mid x)$ or predictive entropy. Using disjoint unlabeled splits from $\mathcal{U}$ (sample splitting or cross-fitting), we form empirical CDFs of $T$ on true target inputs $x_t$ and on transported inputs $\tilde{x}_t = f(x_s)$:

$$\widehat{F}_T(u) = \frac{1}{m_{\text{tgt}}} \sum_{j=1}^{m_{\text{tgt}}} \mathbb{I}\{T(x_t^{(j)}) \leq u\}, \tag{7}$$

$$\widehat{\widetilde{F}}_T(u) = \frac{1}{m_{\text{trans}}} \sum_{j=1}^{m_{\text{trans}}} \mathbb{I}\{T(\tilde{x}_t^{(j)}) \leq u\}. \tag{8}$$

We use the *one-sided* Kolmogorov–Smirnov gap

$$\hat{\delta} = \sup_{u \in \mathbb{R}} \left\{\widehat{\widetilde{F}}_T(u) - \widehat{F}_T(u)\right\}_+, \tag{9}$$

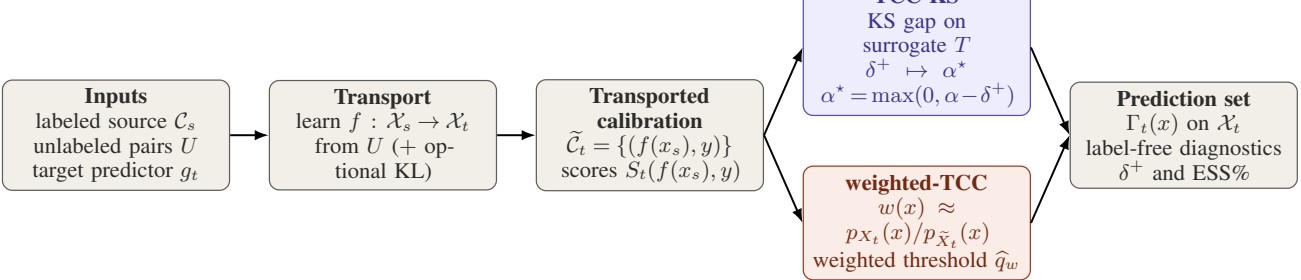

*Figure 2.* Workflow of Transported Conformal Calibration (TCC). Labeled source calibration inputs are mapped into the target input space using a transport map $f$ learned from unlabeled paired observations. The transported examples provide target-space calibration scores, after which TCC corrects residual post-transport mismatch without labeled target calibration data. TCC-KS uses a one-sided KS discrepancy on a label-free surrogate to tighten the internal level to $\alpha^\star = \max(0, \alpha - \delta^+)$, while weighted-TCC reweights transported calibration scores toward the target input distribution. The label-free diagnostics $\delta^+$ and ESS% indicate whether the KS safeguard or post-transport reweighting is more appropriate at deployment.

and inflate it by a DKW bound: with probability at least $1 - \eta$,

$$\delta_T \le \hat{\delta} + \sqrt{\frac{\log(4/\eta)}{2m_{\text{tgt}}}} + \sqrt{\frac{\log(4/\eta)}{2m_{\text{trans}}}} =: \delta^+, \quad (10)$$

where $\delta_T := \sup_u \{\tilde{F}_T(u) - F_T(u)\}_+$ and $F_T, \tilde{F}_T$ are the population CDFs of $T(X_t)$ and $T(\tilde{X}_t)$.

TCC-KS converts $\delta^+$ into a tightened internal level

$$\alpha^\star = \max\{0, \alpha - \delta^+\}, \quad (11)$$

replaces $\alpha$ by $\alpha^\star$ in (4), and outputs

$$\Gamma_t^{\text{TCC-KS}}(x) = \{y \in \mathcal{Y} : S_t(x,y) \le \hat{q}_{1-\alpha^\star}\}. \quad (12)$$

If $\delta^+$ is small, then $\alpha^\star \approx \alpha$ and TCC-KS is tight. If $\delta^+$ is large, $\alpha^\star$ decreases toward 0 and the sets become conservative. This clipping is intentional: when the certified mismatch exceeds $\alpha$, no nonnegative internal level can (in general) certify miscoverage $\le \alpha$ from surrogate information alone. [1]

TCC-KS targets *under-coverage* arising when the target domain is "harder" than the transported one. For uncertainty-increasing surrogates, this corresponds to $T(X_t)$ shifting to larger values than $T(\tilde{X}_t)$, which is captured by $\sup_u(\tilde{F}_T(u) - F_T(u))_+$. The opposite deviation corresponds to transported calibration being conservative and does not need correction.

---

[1]All empirical conformal quantiles in Equations 4 and 12 use the split-conformal order-statistic convention $\hat{q}_{1-\alpha} = S_{(\lceil (n+1)(1-\alpha) \rceil)}$ at the relevant level $\alpha$, whenever this index is at most $n$. If the index is $n+1$, in particular when the relevant level is $\alpha^\star = 0$, we use $\hat{q} = S_{(n)}$ instead of the vacuous $+\infty$ threshold. The resulting finite-sample coverage bound differs only by the usual $1/(n+1)$ conformal discreteness term, which is negligible for our calibration sizes.

### 3.4. Weighted-TCC: Importance-Weighted Correction

weighted-TCC uses the transported labeled calibration scores as in (4), but corrects residual mismatch by reweighting these scores toward the *real* target input distribution. Unlike standard covariate-shift conformal approaches that reweight source samples directly (Tibshirani et al., 2019; Shimodaira, 2000; Sugiyama et al., 2007), weighted-TCC performs *transport-then-reweight*: it treats $\tilde{X}_t = f(X_s)$ as the labeled calibration distribution and $X_t$ as deployment, estimating residual weights

$$w(x) \approx \frac{p_{X_t}(x)}{p_{\tilde{X}_t}(x)}$$

from unlabeled samples $\{x_t^{(j)}\}$ and $\{\tilde{x}_t^{(j)}\}$ (e.g., via a domain classifier), and applying importance-weighted split conformal calibration to obtain a weighted threshold and sets $\Gamma_t^{\text{wTCC}}$. Since the residual shift between $\tilde{X}_t$ and $X_t$ can be smaller than the original source–target shift, reweighting after transport can be substantially more effective. In practice, performance depends on weight stability, so we report label-free diagnostics such as effective sample size (ESS) in Section 4.

**Predictive Alignment during Transport Learning.** Because $\delta^+$ depends on how well transported inputs match the target model's uncertainty profile, it can help to learn $f$ in a prediction-preserving manner. Alongside a reconstruction-style objective on paired data, we consider the label-free regularizer

$$\mathcal{L}_{\text{KL}} = \mathbb{E}_{(x_s, x_t) \sim \mathcal{U}} \left[ \text{KL}\big( \hat{p}_t(\cdot \mid f(x_s)) \,\|\, \hat{p}_t(\cdot \mid x_t) \big) \right], \quad (13)$$

where $\hat{p}_t(\cdot \mid x) = \text{softmax}(g_t(x))$. Empirically, this reduces $\delta^+$ and improves efficiency for both TCC-KS and weighted-TCC.

Figure 2 summarizes the resulting TCC pipeline: paired unlabeled data are used to learn a source-to-target transport

map, transported labeled examples provide target-space calibration scores, and the residual post-transport mismatch is corrected using either the KS-based safeguard or post-transport importance weighting.

## 3.5. Theoretical Guarantees

We now state a high-probability target coverage guarantee for TCC-KS under an approximate condition that relates surrogate mismatch to score mismatch, an oracle-weight validity statement for weighted-TCC, and an impossibility result showing that, in the absence of target labels, nontrivial distribution-free guarantees necessarily require additional structure. Proofs and supporting lemmas are in Appendix A.

### 3.5.1. AN IMPOSSIBILITY RESULT

**Theorem 3.1** (Impossibility of nontrivial label-free validity without structural assumptions). *Assume $|\mathcal{Y}| \geq 2$ and fix any $\alpha \in (0,1)$. Consider any (possibly randomized) procedure that, given (i) a labeled transported calibration sample $\tilde{\mathcal{C}}_t$ and (ii) unlabeled target inputs (and any unlabeled transported inputs), outputs prediction sets $\Gamma_t : \mathcal{X}_t \to 2^{\mathcal{Y}}$ without observing any labeled target examples. Suppose the procedure is required to satisfy the marginal coverage guarantee in Equation* (1) *uniformly over all data-generating distributions $P$ whose target marginal $P_{X_t}$ is fixed. Then*

$$\sup_{P:\, P_{X_t} \text{ fixed}} \mathbb{E}\big[\, |\Gamma_t(X_t)|\, \big] \; \geq \; (1-\alpha)\,|\mathcal{Y}|. \quad (14)$$

**Implication.** Theorem 3.1 shows that, without labeled target data, distribution-free marginal validity alone cannot yield informative prediction sets in the worst case: any uniformly valid label-free procedure must be *near-vacuous* (e.g., with worst-case expected set size at least $(1-\alpha)|\mathcal{Y}|$). Consequently, any method that aims to be nontrivial must assume *some* relationship between observable quantities (computable from unlabeled inputs and model outputs) and the unobservable true-label score distribution $S_t(X_t, Y)$. Alternative approaches, such as assuming covariate shift for importance weighting, perfect calibration, conditional independence, or latent exchangeability, do not eliminate assumptions. Instead, they replace them with different structural conditions. Our contribution is to make this requirement explicit via Assumption A2, to provide label-free diagnostics (e.g., $\delta^+$) that assess when it is plausible, and to design a procedure that degrades gracefully when the diagnostics indicate mismatch.

**Coverage Guarantees under Approximate Surrogate Control.** Given Theorem 3.1, we now formalize a sufficient condition for valid calibration transfer and derive finite-sample guarantees.

**Assumptions.** **A1 (Task Invariance).** Paired views share the same label semantics across $\mathcal{X}_s$ and $\mathcal{X}_t$.

**A2 (Approximate Surrogate Control).** Let $F_S(u) = \mathbb{P}(S_t(X_t, Y) \leq u)$ and $\tilde{F}_S(u) = \mathbb{P}(S_t(\tilde{X}_t, Y) \leq u)$, and let $F_T, \tilde{F}_T$ denote the marginal CDFs of $T(X_t)$ and $T(\tilde{X}_t)$. Define the one-sided gaps

$$\delta_S = \sup_{u \in \mathbb{R}} \big\{ \tilde{F}_S(u) - F_S(u) \big\}_+,$$

and

$$\delta_T = \sup_{u \in \mathbb{R}} \big\{ \tilde{F}_T(u) - F_T(u) \big\}_+.$$

We assume $\delta_S \leq \delta_T + \varepsilon$ for some $\varepsilon \geq 0$.

**A3 (Sample Usage).** The samples used to learn $f$, to compute transported quantiles, and to estimate $\hat{\delta}$ are independent via sample splitting or cross-fitting.

**Theorem 3.2** (TCC-KS coverage under approximate surrogate control). *Assume A1, A2, and A3. Let $\delta^+$ be defined by* (10), $\alpha^\star$ *by* (11), *and $\Gamma_t^{\text{TCC-KS}}$ by* (12). *Then for any fixed $\eta \in (0,1)$, with probability at least $1-\eta$ over the unlabeled samples used to compute $\delta^+$,*

$$\mathbb{P}\big\{ Y \in \Gamma_t^{\text{TCC-KS}}(X_t) \big\} \; \geq \; 1 - \big(\alpha^\star + \delta^+ + \varepsilon\big),$$

*where the probability is over both the transported labeled calibration sample used to form $\hat{q}_{1-\alpha^\star}$ (equivalently, $\tilde{\mathcal{C}}_t$) and the test draw $(X_t, Y)$. In particular, whenever $\delta^+ \leq \alpha$ (so that $\alpha^\star + \delta^+ = \alpha$), we obtain*

$$\mathbb{P}\{ Y \in \Gamma_t^{\text{TCC-KS}}(X_t) \} \geq 1 - (\alpha + \varepsilon).$$

**A Guarantee for Weighted-TCC (Oracle Weights).** Validity follows from standard importance-weighted conformal arguments applied *after* transport.

**Proposition 3.3** (Oracle-weight validity of weighted-TCC). *Assume (i) $\tilde{X}_t$ and $X_t$ satisfy covariate shift, meaning that $\mathbb{P}(Y \mid \tilde{X}_t = x) = \mathbb{P}(Y \mid X_t = x)$ for all $x$, and (ii) the density ratio $w(x) = p_{X_t}(x)/p_{\tilde{X}_t}(x)$ exists and is used in an importance-weighted split-conformal calibration procedure on the transported labeled scores. Then the resulting sets $\Gamma_t^{\text{weighted}-\text{TCC}}$ satisfy target-domain marginal coverage*

$$\mathbb{P}\big\{ Y \in \Gamma_t^{\text{weighted}-\text{TCC}}(X_t) \big\} \; \geq \; 1 - \alpha.$$

**Interpretation.** TCC-KS exposes a label-free mismatch certificate $\delta^+$ that controls the adjustment $\alpha^\star$ and serves as a deployment signal: small $\delta^+$ yields tight sets, while large $\delta^+$ triggers conservative behavior. Assumption A2 formalizes a proxy condition under which observable mismatch in $T(X)$ upper bounds score-relevant mismatch, enabling nontrivial label-free guarantees in the sense ruled out by Theorem 3.1 absent additional structure. Weighted-TCC is complementary: it can be more efficient when the residual weights are stable (high ESS), but may be unreliable when weights concentrate.

# 4. Evaluation

**Goals.** We evaluate **TCC-KS** and **weighted-TCC** for label-free conformal prediction under paired cross-space shift. All reported metrics are computed on a held-out labeled *target test* set; target labels are used only for evaluation.

**Primary Metrics.** We report empirical target marginal coverage and average prediction set size. These summarize, respectively, validity and efficiency of the prediction sets on the target domain.

**Method-Aligned Diagnostics.** To connect empirical behavior to our two correction mechanisms, we also report diagnostics when relevant. For **TCC-KS**, we report the one-sided KS inflation term $\delta^+$ and the induced adjusted level $\alpha^\star$ which determines the internal calibration quantile used by TCC-KS. Empirically, smaller $\delta^+$ should correspond to operating closer to a near-nominal regime, while larger $\delta^+$ triggers more conservative sets.

For **weighted-TCC**, we report a weight-stability diagnostic computed on the unlabeled sample used for weight estimation,

$$\text{ESS} = \frac{\left(\sum_{j=1}^{N} w_j\right)^2}{\sum_{j=1}^{N} w_j^2}, \qquad \text{ESS\%} = 100 \cdot \frac{\text{ESS}}{N},$$

where larger ESS% indicates less concentrated weights and typically more reliable post-transport reweighting.

**Datasets and Paired Shift.** We use two *graded-severity* corruption benchmarks and one multi-sensor dataset with naturally paired modalities. On **CIFAR-100-C** and **Tiny-ImageNet-C** (Hendrycks & Dietterich, 2019), clean images are sources $x_s$ and corrupted images are targets $x_t$, yielding paired samples $(x_s, x_t)$ with identical labels. Varying corruption *type* and *severity* provides a controlled continuum from mild covariate shift to near support-mismatch regimes at high severity, where overlap and importance weighting can degrade. Unless stated otherwise, we average over four corruptions (contrast, brightness, shot noise, motion blur) and severities 1–5, totaling 20 target conditions. On **SEN12MS (SAR→RGB)** (Schmitt et al., 2019), SAR is $x_s$ and RGB is $x_t$ using co-registered tiles, representing a modality shift with typically lower overlap than corruptions. Furthermore, we present additional results on SUN RGB-D (Song et al., 2015) in Appendix E.

**Models and Protocol.** We fix a target classifier $g_t$ per dataset across all methods. We learn a transport map $f : \mathcal{X}_s \to \mathcal{X}_t$ from unlabeled paired data using either no KL (reconstruction) or +KL (reconstruction plus a label-free predictive KL alignment term matching $g_t(x_t)$ to $g_t(f(x_s))$ on paired inputs). We use the LAC score $S_t(x,y) = 1 - \hat{p}_t(y \mid x)$ and the surrogate $T(x) = 1 - \max_y \hat{p}_t(y \mid x)$. To satisfy Assumption A3, we split unlabeled pairs into

| Dataset | Method | $\alpha = 0.1$ | $\alpha = 0.2$ |
|---|---|---|---|
| CIFAR-100-C | Oracle CP | 0.89 (37.63) | 0.79 (23.70) |
| | Transported CP | 0.89 (37.18) | 0.79 (23.33) |
| | TCC-KS | 0.92 (44.35) | 0.82 (26.68) |
| | weighted-TCC | 0.89 (37.15) | 0.79 (23.23) |
| | WCP | 0.92 (43.90) | 0.83 (28.89) |
| Tiny-ImageNet-C | Oracle CP | 0.90 (69.68) | 0.80 (42.70) |
| | Transported CP | 0.90 (70.16) | 0.80 (43.05) |
| | TCC-KS | 0.93 (86.33) | 0.83 (50.58) |
| | weighted-TCC | 0.90 (69.71) | 0.80 (42.60) |
| | WCP | 0.93 (88.54) | 0.85 (57.48) |
| SAR→RGB | Oracle CP | 0.89 (1.02) | 0.79 (0.83) |
| | Transported CP | 0.90 (1.04) | 0.85 (0.93) |
| | TCC-KS | 0.99 (2.51) | 0.99 (2.51) |
| | weighted-TCC | 0.90 (1.03) | 0.84 (0.91) |
| | WCP | 1.00 (2.87) | 0.99 (2.63) |

*Table 1.* Corrupted-image results average over 20 target conditions. Entries report coverage with set size in parentheses. Per-condition distributions and detailed breakdowns are reported in Appendix C.

| Dataset | Method | $\alpha = 0.1$ | $\alpha = 0.2$ |
|---|---|---|---|
| CIFAR-100-C | Oracle CP | 0.89 (42.11) | 0.79 (27.54) |
| | Transported CP | 0.88 (41.05) | 0.78 (26.52) |
| | TCC-KS | 0.91 (47.66) | 0.81 (29.67) |
| | weighted-TCC | 0.88 (40.94) | 0.78 (26.13) |
| | WCP | 0.93 (53.10) | 0.86 (37.22) |
| Tiny-ImageNet-C | Oracle CP | 0.899 (77.47) | 0.80 (48.72) |
| | Transported CP | 0.898 (77.17) | 0.79 (47.86) |
| | TCC-KS | 0.930 (91.75) | 0.82 (55.07) |
| | weighted-TCC | 0.897 (76.84) | 0.79 (47.45) |
| | WCP | 0.965 (120.26) | 0.91 (86.73) |

*Table 2.* **Stress test.** Shot noise and motion blur at severities 4–5. Coverage with set size in parentheses.

disjoint subsets for (i) learning the transport map, (ii) estimating the KS discrepancy, and (iii) estimating residual importance weights, or we use cross-fitting. Main-text results report $\alpha \in \{0.1, 0.2\}$. Appendix B provides complete implementation details (data splits, architectures, and baselines), Appendix C and Appendix D report extended results for Experiments 1 and 2, and Appendix F reports an additional surrogate-sensitivity study using predictive entropy. Appendix G provides additional diagnostic audits and stress-case documentation, and Appendix G.5 summarizes an operational procedure based on $\delta^+$ and ESS%.

**Methods Compared.** We compare **Oracle CP** (target-labeled calibration, reference only), **Transported CP** (split conformal on transported labeled scores $S_t(f(x_s), y)$), **TCC-KS** (Transported CP with KS inflation $\delta^+$), **weighted-TCC** (post-transport importance weighting between $\tilde{X}_t = f(X_s)$ and $X_t$), and **WCP (no transport)** (weighted conformal without transport).

## 4.1. Overall Performance and a Severe-Shift Stress Test

**Description.** We report marginal coverage and average set size at $\alpha \in \{0.1, 0.2\}$. We first average over the 20 target conditions on CIFAR-100-C and Tiny-ImageNet-C and report SAR→RGB on its paired evaluation split (Table 1). We then report a stress test on shot noise and motion blur at severities 4–5 (Table 2).

**Results.** On the overall averages (Table 1), Transported CP and weighted-TCC are close to nominal on CIFAR-100-C and Tiny-ImageNet-C, suggesting that transport often reduces the residual mismatch to a mild regime. TCC-KS typically achieves a safety margin through its $\delta^+$-based adjustment, at the cost of moderate set-size increases. WCP (no transport) also attains high coverage, but with larger sets than transport-based calibration, indicating lower efficiency. On SAR→RGB, transported calibration remains effective on average. TCC-KS attains near-perfect coverage while remaining non-vacuous (average size $\approx 2.5$), and WCP is slightly larger.

The stress test (Table 2) separates the methods more clearly. Transported CP and weighted-TCC exhibit mild under-coverage at $\alpha = 0.1$, whereas TCC-KS restores coverage with a moderate increase in set size. WCP remains more conservative but with substantially larger sets, highlighting the benefit of correcting after transport. Additional results across corruptions and severities are provided in Appendix C.

## 4.2. Effect of Transport Quality

**Description.** We test how calibration transfer depends on the quality of the learned transport map $f$ and connect the behavior to both correction mechanisms. For each dataset we train $f$ using `no KL` vs `+KL` and compare an early and late checkpoint (epochs 5 and 150) at $\alpha = 0.1$. We recompute Transported CP, TCC-KS, and weighted-TCC. To expose mechanisms, we report the unlabeled KS inflation $\delta^+$ and a weight-stability diagnostic for weighted-TCC, the effective sample size of residual weights as a percentage over $N = 10,000$ unlabeled points used for weight estimation, ESS% := $100 \cdot \text{ESS}/N$. Full sweeps and variability appear in Appendix D.

**Results.** Table 3 shows that as transport improves, $\delta^+$ decreases, reducing the KS-based tightening and shrinking TCC-KS sets while preserving conservativeness. In parallel, ESS% increases, indicating more stable post-transport weights and a regime where weighted-TCC is most credible.

On Tiny-ImageNet-C, `+KL` substantially reduces mismatch and stabilizes weights at epoch 150, with $\delta^+$ dropping from 0.1006 to 0.0148 and ESS% rising from 48.4 to 91.8. In this regime, weighted-TCC is both efficient and near nominal (0.900 coverage with size 69.71), while TCC-KS remains conservative with larger sets. On CIFAR-100-C, `+KL` yields higher ESS% and a markedly smaller $\delta^+$ (from 0.0521 to 0.0152 at epoch 150), and weighted-TCC matches Transported CP efficiency at epoch 150 while TCC-KS maintains a safety margin when uncorrected coverage is mildly low (0.894). On SAR→RGB, $\delta^+$ remains large at epoch 150 under `+KL` (0.3200), and TCC-KS continues to act as a guardrail with high coverage, while Transported CP and

weighted-TCC can under-cover, consistent with persistent residual mismatch. We provide results for further transport model configurations in Appendix D.

## 4.3. Assumption Validation and Stress Regimes

**Description.** TCC-KS relies on a deployable, label-free mismatch certificate $\delta^+$ computed from the surrogate $T$, while target validity depends on the true-label score distribution. Using held-out target labels *only for evaluation*, we stress-test this mechanism across 328 configurations spanning transport checkpoints, two transport objectives ($\ell_1$ and $\ell_1$+KL), four corruptions at five severities on CIFAR-100-C and Tiny-ImageNet-C, plus SAR→RGB. This sweep lets us (i) identify regimes where transport-only calibration can under-cover, (ii) validate $\delta^+$ (and ESS% for weighting) as actionable deployment signals, and (iii) quantify how closely $\delta^+$ tracks evaluation-only score-side discrepancies.

Table 4 shows three representative cases. Our central findings are: **(1) Under-coverage recovery.** Across 328 configurations, we identify 11 cases (3.4%) where transport-only under-covers by $> 1$pp (mean 87.9% vs. oracle 89.9%). TCC-KS restores valid coverage in all 11 cases (100% success rate), achieving mean 97.8% coverage with $1.82\times$ inflation. **(2) Diagnostic validation.** The unlabeled $\delta^+$ correlates 0.772 with set-size inflation and 0.734 with safety margin, supporting it as an actionable deployment signal. **(3) Regime structure.** Most settings fall in a mild regime, while a smaller fraction exhibit moderate-to-extreme mismatch where TCC-KS becomes more conservative. A complete documentation of the cases appears in Appendix G.

**Surrogate–Score Coupling and Empirical Slack.** To audit whether the label-free certificate $\delta^+$ is systematically optimistic relative to true-label score discrepancies, we evaluate at epoch 150. In Table 5, we report: (i) coupling $\rho(T, S)$ between $T$ and the true-label score $S_t(\cdot, Y)$; (ii) an evaluation-only one-sided score discrepancy diagnostic $\delta_S^+$ (requires target labels); and (iii) an evaluation-only *slack proxy* $\widehat{\varepsilon}_{\text{eval}} := \max\{0, \delta_S^+ - \delta^+\}$, summarized as (med / max) over settings. Intuitively, $\widehat{\varepsilon}_{\text{eval}}$ measures how often (and by how much) the certificate $\delta^+$ could understate the observed score-side discrepancy in our evaluation suite.

The results support two deployment-relevant conclusions. First, in target mode the chosen surrogate is a meaningful proxy for difficulty (high $\rho(T, S)$ and stable bin behavior), motivating its use for label-free monitoring. Second, the certificate $\delta^+$ is *not systematically optimistic* relative to the observed score-side discrepancy on corruption benchmarks: in the mild/certified regimes on CIFAR/Tiny-ImageNet-C (all rows have $\delta^+ < \alpha_{\text{nom}} = 0.1$ at epoch 150), the median slack proxy is at most $9.4 \times 10^{-3}$ and is often 0, while worst-case slack remains bounded ($\leq 4.44 \times 10^{-2}$ in our suite). The largest slack occurs for CIFAR-100-C with

| Dataset | Objective | Epoch | $\delta^+$ | ESS% | Trans. CP (cov, size) | TCC-KS (cov, size) | weighted-TCC (cov, size) |
|---|---|---|---|---|---|---|---|
| CIFAR-100-C | `no KL` | 5 | 0.1605 | 37.4 | 0.954 (54.43) | 0.975 (65.20) | 0.939 (48.42) |
| | `no KL` | 150 | 0.0521 | 72.9 | 0.920 (44.03) | 0.942 (49.76) | 0.910 (40.71) |
| | `+KL` | 5 | 0.0434 | 79.5 | 0.905 (39.33) | 0.960 (61.21) | 0.903 (38.87) |
| | `+KL` | 150 | 0.0152 | 93.3 | 0.894 (37.18) | 0.927 (44.35) | 0.894 (37.15) |
| Tiny-ImageNet-C | `no KL` | 5 | 0.3422 | 13.0 | 0.982 (146.47) | 1.000 (194.06) | 0.976 (132.06) |
| | `no KL` | 150 | 0.1006 | 48.4 | 0.931 (91.80) | 0.964 (118.96) | 0.923 (83.61) |
| | `+KL` | 5 | 0.0965 | 54.4 | 0.923 (81.04) | 0.981 (150.93) | 0.916 (78.14) |
| | `+KL` | 150 | 0.0148 | 91.8 | 0.901 (70.16) | 0.935 (86.33) | 0.900 (69.71) |
| SAR→RGB | `no KL` | 5 | 0.7405 | 26.0 | 1.000 (4.00) | 1.000 (4.00) | 1.000 (3.82) |
| | `no KL` | 150 | 0.4947 | 37.0 | 1.000 (2.68) | 1.000 (3.76) | 1.000 (2.56) |
| | `+KL` | 5 | 0.7782 | 35.3 | 0.980 (1.40) | 0.991 (1.70) | 0.979 (1.40) |
| | `+KL` | 150 | 0.3200 | 39.5 | 0.890 (1.04) | 0.999 (2.51) | 0.897 (1.03) |

*Table 3.* **Transport-quality sweep** ($\alpha = 0.1$). We report $\delta^+$ (TCC-KS) and ESS% (weighted-TCC) with coverage and set size.

| Case | Example Setting | $\delta^+$ | Transport Cov / Size | TCC-KS Cov / Size |
|---|---|---|---|---|
| Mild | CIFAR-100, contrast, s3, $\ell_1$+KL, e150 | 0.023 | 0.905 / 32.6 | 0.931 / 38.7 (1.26×) |
| Under-coverage | Tiny-ImageNet, mblur, s2, $\ell_1$, e20 | 0.036 | **0.875** / 68.5 **(-2.7pp)** | **0.984** / 136.6 **(1.76×)** |
| Extreme | SAR→RGB, $\ell_1$+KL, e5 | 0.778 | 0.979 / 1.4 | 0.990 / 1.7 (1.66×) |

*Table 4.* **Representative stress regimes.** Row 1: Mild mismatch ($\delta^+$ small) yields modest conservativeness. Row 2: Transport-only under-covers; TCC-KS detects via $\delta^+$ and restores a safety margin. Row 3: Extreme mismatch triggers conservative behavior.

$\ell_1$+KL, indicating a harder regime where certificate tightness can degrade; importantly, these are exactly the regimes where TCC-KS becomes more conservative, aligning with its intended role as a safety guardrail.

Finally, SAR→RGB illustrates a distinct cross-modal stress regime: $\delta^+$ is large (well above $\alpha_{\mathrm{nom}}$) and TCC-KS tightens to $\alpha^\star = 0$, yielding conservative behavior. In this regime the slack proxy is zero only because $\delta^+$ is already large. The takeaway is that the label-free certificate flags the regime as high-mismatch, consistent with the observed conservativeness and with our stress-case taxonomy in Table 4.

## 5. Discussion

We view conformal calibration as a statistical asset that can be *transferred* across domains when target labels are scarce but unlabeled pairing is available. TCC makes this transfer explicit via two steps: (i) learn a transport map $f$ from paired unlabeled data to move labeled source calibration examples into the target input space, and (ii) apply a target-label-free correction to account for residual mismatch between transported inputs $f(X_s)$ and true target inputs $X_t$. This decomposition yields two complementary correction mechanisms. **TCC-KS** is a validity-oriented guardrail: it uses an unlabeled one-sided KS discrepancy on a surrogate uncertainty statistic $T$ to form a mismatch certificate $\delta^+$ and an adjusted internal level $\alpha^\star = \max(0, \alpha - \delta^+)$.

When mismatch is mild ($\delta^+$ small), TCC-KS stays close to transported split conformal; as mismatch grows, $\alpha^\star$ shrinks and the procedure increases coverage protection in a controlled way. Theorem 3.2 formalizes this behavior via a high-probability target coverage bound driven by $\delta^+$, and highlights the near-nominal regime when $\delta^+ \leq \alpha$. Thus, the correction layer is most beneficial when transport leaves non-negligible residual mismatch: transported split conformal may then be efficient but no longer target-valid, while TCC-KS trades modestly larger sets for an explicit target-domain coverage safeguard.

**weighted-TCC** is an efficiency-oriented complement: it reweights calibration scores *after* transport, targeting the residual shift between $\tilde{X}_t$ and $X_t$ rather than the original source–target gap. In regimes where the residual shift is well-approximated by covariate shift and the estimated weights are stable, transport-then-reweight can substantially tighten sets, approaching target-calibrated efficiency without labeled target calibration data. Together, $\delta^+$ and ESS explain when each correction is likely to be reliable, consistent with the diagnostic audits in Section 4.

**Diagnostics as Deployment Signals.** A practical feature of TCC is that its key operating signals are label-free. For TCC-KS, $\delta^+$ summarizes residual mismatch in the target model's uncertainty profile and directly controls the safeguard strength; for weighted-TCC, ESS indicates when

| Dataset | Mode | $\rho(T, S)\uparrow$ | ESS%$\uparrow$ | $\delta^+\downarrow$ | $\delta_S^+$ (med/max)$\downarrow$ | $\widehat{\varepsilon}_{\text{eval}}$ (med/max)$\downarrow$ |
|---|---|---|---|---|---|---|
| CIFAR-100-C | target | 0.310 | — | — | — | — |
| | transport (no KL) | 0.254 | 72.9 | 0.0256 | 0.027 / 0.045 | 0.0014 / 0.0194 |
| | transport (+KL) | 0.311 | 93.3 | 0.0306 | 0.040 / 0.075 | 0.0094 / 0.0444 |
| Tiny-ImageNet-C | target | 0.325 | — | — | — | — |
| | transport (no KL) | 0.239 | 48.4 | 0.0493 | 0.020 / 0.080 | 0.0000 / 0.0307 |
| | transport (+KL) | 0.321 | 91.8 | 0.0345 | 0.037 / 0.061 | 0.0025 / 0.0265 |
| SAR$\rightarrow$RGB | target | 0.973 | — | — | — | — |
| | transport (no KL) | 0.120 | 35.3 | 0.1889 | 0.086 / 0.086 | 0.0000 / 0.0000 |
| | transport (+KL) | 0.966 | 39.5 | 0.3549 | 0.156 / 0.156 | 0.0000 / 0.0000 |

*Table 5.* **Surrogate–score coupling, label-free mismatch, and empirical slack (epoch 150).** Target mode confirms $T$ is a meaningful difficulty proxy ($\rho(T, S)\approx 0.3$ on CIFAR/Tiny-ImageNet-C and 0.973 on SAR$\rightarrow$RGB). Transport mode reports weight stability (ESS%), the deployable certificate $\delta^+$, the evaluation-only score discrepancy diagnostic $\delta_S^+$, and the evaluation-only slack proxy $\widehat{\varepsilon}_{\text{eval}}$.

reweighting is likely to be stable. These diagnostics support simple operational choices: use TCC-KS as a default when validity is paramount, and consider weighted-TCC when ESS is sufficiently high and tighter sets are desired (Appendix G.5). As a sensitivity check (Appendix F), replacing least-confidence with predictive entropy yields similar qualitative trends in $\delta^+$ and downstream behavior on our benchmark suite.

**Scope and Limitations.** Our setting assumes access to unlabeled paired observations linking source and target inputs, as in sensor upgrades, synchronized multi-view systems, and paired remote sensing products. Imperfect or approximate pairing primarily affects TCC through transport quality: weaker or noisier correspondences can degrade $f$, increase the residual mismatch between $\widetilde{X}_t = f(X_s)$ and $X_t$, and therefore lead TCC-KS to produce more conservative sets or weighted-TCC to rely on less stable weights.

As in standard conformal prediction, our guarantees are marginal; group- or stratified-coverage objectives can be incorporated with standard techniques, while stronger conditional guarantees remain complementary. TCC-KS depends on a surrogate uncertainty statistic: when residual mismatch affects this statistic, the certificate tightens $\alpha^\star$ to protect coverage; shifts that do not register in such label-free signals are intrinsically difficult to detect without target labels. Finally, our finite-sample analysis uses explicit sample splitting among transport learning, calibration, and mismatch estimation, which can reduce effective sample size when unlabeled budgets are small but becomes less restrictive as paired data increase.

## 6. Conclusion

We introduced *Transported Conformal Calibration (TCC)* for conformal calibration transfer: labeled calibration data exist only in a source space, but prediction sets are required in a target space connected through unlabeled paired observations. TCC learns a transport map from paired data, calibrates on transported labeled examples, and then cor-

rects residual post-transport mismatch using only unlabeled target inputs. We instantiated this correction in two complementary ways: **TCC-KS**, a conservative guardrail driven by a label-free mismatch certificate $\delta^+$, and **weighted-TCC**, an efficiency-oriented post-transport reweighting method guided by weight-stability diagnostics. Across corrupted-image benchmarks and a cross-modal remote-sensing setting, TCC provides reliable label-free coverage transfer across shift severities and surfaces actionable signals for deployment. We hope this framework broadens the applicability of conformal prediction to real transitions where labels are expensive but paired unlabeled data are available.

## Impact Statement

This work contributes to reliable machine learning by extending conformal calibration to settings where labeled calibration data are available in one input space, while predictions are needed in another. Such situations arise whenever data representations, modalities, domains, sensors, or processing pipelines change, and obtaining fresh labeled calibration data in the deployment setting is costly, delayed, or unavailable.

By using unlabeled paired observations to transfer calibration, the proposed framework broadens the range of settings in which uncertainty-aware prediction sets can be constructed without target-domain labels. The label-free diagnostics introduced in this work also provide practical signals about when transfer is reliable and when more conservative prediction sets are appropriate. In high-stakes deployments, these diagnostics are best used together with domain expertise, monitoring, and targeted labeled auditing when available.

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

# Appendix

This appendix contains proofs, implementation details, and extended experimental documentation related to the paper.

- Appendix A contains proofs of Theorems 3.1 and 3.2 and Proposition 3.3, along with supporting lemmas (including the one-sided DKW bound, a sufficient condition with $\varepsilon=0$, and a quantile inflation lemma).

- Appendix B details the full experimental protocol and diagnostics.

- Appendix E reports additional results on the paired RGB-D scene-classification benchmark SUN RGB-D (Song et al., 2015).

- Appendix F reports a surrogate sensitivity check using predictive entropy.

- Appendix G documents the full 328-configuration audit of Assumption A2 violations and operational guidance (Appendix G.5).

## A. Proofs and Supporting Lemmas

This appendix provides proofs for Theorem 3.1, Theorem 3.2, and Proposition 3.3, along with supporting lemmas. Throughout, $F_S$ and $\tilde{F}_S$ denote the CDFs of the true-label scores $S_t(X_t, Y)$ and $S_t(\tilde{X}_t, Y)$, while $F_T$ and $\tilde{F}_T$ denote the CDFs of the label-free surrogate $T(X_t)$ and $T(\tilde{X}_t)$, with $\tilde{X}_t = f(X_s)$.

### A.1. Proof of Theorem 3.1

Theorem 3.1 formalizes that, without labeled target data and without structural assumptions coupling observable quantities to the unobservable true-label score distribution, distribution-free target validity forces prediction sets to be essentially vacuous in the worst case. The key obstruction is that, from unlabeled target inputs alone, one cannot rule out target conditionals $P(Y \mid X_t)$ that are arbitrary.

[Theorem 3.1] Assume $|\mathcal{Y}| = K \geq 2$ and fix any $\alpha \in (0, 1)$. Consider any (possibly randomized) procedure that, given (i) a labeled transported calibration sample $\widetilde{\mathcal{C}}_t$ and (ii) unlabeled target inputs (and any unlabeled transported inputs), outputs prediction sets $\Gamma_t : \mathcal{X}_t \to 2^{\mathcal{Y}}$ without observing any labeled target examples. Suppose the procedure is required to satisfy the marginal coverage guarantee (1) *uniformly* over all data-generating distributions $P$ whose target marginal $P_{X_t}$ is fixed. Then, for the subclass of such distributions where $Y$ is conditionally independent of $X_t$ and uniform on $\mathcal{Y}$, the procedure must satisfy

$$\sup_{P : P_{X_t} \text{ fixed}} \mathbb{E}\big[|\Gamma_t(X_t)|\big] \geq (1 - \alpha)K.$$

In particular, any procedure that outputs substantially smaller-than-$K$ sets on all distributions in this class cannot guarantee (1) without additional assumptions.

*Proof.* Fix an arbitrary target marginal distribution $P_{X_t}$ on $\mathcal{X}_t$ (this is the "fixed" target marginal in the theorem statement) and let $K = |\mathcal{Y}|$. Consider the following data-generating distribution $P^\circ$ on $(X_s, X_t, Y)$:

- $X_t \sim P_{X_t}$,

- $Y \sim \text{Unif}(\mathcal{Y})$ and $Y$ is independent of $X_t$,

- $(X_s, \tilde{X}_t)$ are generated independently of $(X_t, Y)$ in any way that yields a well-defined transported labeled calibration sample $\widetilde{\mathcal{C}}_t = \{(\tilde{x}_t^{(i)}, y^{(i)})\}_{i=1}^n$ (e.g., one may take $X_s = \tilde{X}_t$ and $f$ to be the identity on $\mathcal{X}_t$).

This construction ensures that the procedure receives exactly the same type of inputs as in the main paper (a labeled transported calibration sample and unlabeled target inputs), but the target label $Y$ at test time carries no information beyond being uniform over $\mathcal{Y}$.

Let $\Gamma_t$ be the (possibly random) prediction-set function output by the procedure after observing its input data. Draw an independent test pair $(X_t, Y) \sim P^\circ$. Conditional on the procedure's internal randomness and all observed samples (hence conditional on $\Gamma_t$), the test label $Y$ remains uniform on $\mathcal{Y}$ and independent of $X_t$ and $\Gamma_t(X_t)$. Therefore,

$$\mathbb{P}\big(Y \in \Gamma_t(X_t) \mid \Gamma_t, X_t\big) \;=\; \frac{|\Gamma_t(X_t)|}{K}.$$

Taking expectations yields

$$\mathbb{P}\big(Y \in \Gamma_t(X_t)\big) \;=\; \mathbb{E}\Big[\frac{|\Gamma_t(X_t)|}{K}\Big] \;=\; \frac{1}{K}\,\mathbb{E}\big[|\Gamma_t(X_t)|\big].$$

If the procedure is to satisfy the marginal coverage requirement (1) under $P^\circ$, then

$$\frac{1}{K}\,\mathbb{E}\big[|\Gamma_t(X_t)|\big] \;=\; \mathbb{P}\big(Y \in \Gamma_t(X_t)\big) \;\geq\; 1 - \alpha,$$

which implies

$$\mathbb{E}\big[|\Gamma_t(X_t)|\big] \;\geq\; (1-\alpha)K.$$

Since $P^\circ$ has the prescribed fixed target marginal $P_{X_t}$, this lower bound holds for the worst-case expected set size over the entire class of distributions with that fixed marginal. This proves the claim. $\qquad\square$

**Remark (relation to "vacuous" sets).** Theorem 3.1 shows that, without additional structure relating observables to the unobservable score distribution, any procedure that is distribution-free valid over all $P$ with a fixed target marginal must have *worst-case* expected set size at least $(1-\alpha)|\mathcal{Y}|$, which is near-vacuous when $|\mathcal{Y}|$ is large and $\alpha$ is small. This motivates making explicit proxy assumptions (such as Assumption A2) that enable nontrivial set sizes while retaining coverage guarantees.

### A.2. Concentration for the One-Sided Surrogate Discrepancy

Recall the population one-sided surrogate gap

$$\delta_T \;=\; \sup_{u \in \mathbb{R}}\{\tilde{F}_T(u) - F_T(u)\}_+,$$

and its empirical version

$$\hat{\delta} \;=\; \sup_{u \in \mathbb{R}}\{\widehat{\tilde{F}}_T(u) - \widehat{F}_T(u)\}_+,$$

where $\widehat{F}_T$ is computed from $m_{\text{tgt}}$ unlabeled target inputs and $\widehat{\tilde{F}}_T$ from $m_{\text{trans}}$ transported inputs, as in (7).

**Lemma A.1** (High-probability inflation for the one-sided gap). *Let $\epsilon_{\text{tgt}} = \sqrt{\frac{\log(4/\eta)}{2m_{\text{tgt}}}}$ and $\epsilon_{\text{trans}} = \sqrt{\frac{\log(4/\eta)}{2m_{\text{trans}}}}$. Then, with probability at least $1 - \eta$,*

$$\delta_T \;\leq\; \hat{\delta} \;+\; \epsilon_{\text{tgt}} \;+\; \epsilon_{\text{trans}}.$$

*Proof.* By the DKW inequality, for any $\epsilon > 0$,

$$\mathbb{P}\bigg(\sup_u |F_T(u) - \widehat{F}_T(u)| > \epsilon\bigg) \;\leq\; 2e^{-2m_{\text{tgt}}\epsilon^2}, \qquad \mathbb{P}\bigg(\sup_u |\tilde{F}_T(u) - \widehat{\tilde{F}}_T(u)| > \epsilon\bigg) \;\leq\; 2e^{-2m_{\text{trans}}\epsilon^2}.$$

With $\epsilon_{\text{tgt}} = \sqrt{\frac{\log(4/\eta)}{2m_{\text{tgt}}}}$ and $\epsilon_{\text{trans}} = \sqrt{\frac{\log(4/\eta)}{2m_{\text{trans}}}}$, each event holds with probability at least $1 - \eta/2$. By a union bound, with probability at least $1 - \eta$ we have simultaneously

$$\sup_u |F_T(u) - \widehat{F}_T(u)| \leq \epsilon_{\text{tgt}}, \qquad \sup_u |\tilde{F}_T(u) - \widehat{\tilde{F}}_T(u)| \leq \epsilon_{\text{trans}}.$$

On this event, for all $u$,

$$\tilde{F}_T(u) - F_T(u) \;\leq\; \big(\widehat{\tilde{F}}_T(u) + \epsilon_{\text{trans}}\big) - \big(\widehat{F}_T(u) - \epsilon_{\text{tgt}}\big) = \big(\widehat{\tilde{F}}_T(u) - \widehat{F}_T(u)\big) + \epsilon_{\text{tgt}} + \epsilon_{\text{trans}}.$$

Taking positive parts and then the supremum over $u$ gives the claim. $\qquad\square$

### A.3. A sufficient Condition Implying $\varepsilon = 0$

The main paper uses Assumption A2, which directly relates the one-sided score gap $\delta_S$ to the one-sided surrogate gap $\delta_T$ up to slack $\varepsilon$. For intuition, it is sometimes helpful to recognize a structural sufficient condition under which $\varepsilon = 0$; this subsection is *not* needed to apply TCC-KS.

**A2′ (Score–surrogate alignment; sufficient for $\varepsilon = 0$).** For every $u \in \mathbb{R}$, the function $t \mapsto \mathbb{P}(S_t(X_t, Y) \leq u \mid T(X_t) = t)$ is nonincreasing. Moreover, for all $u$ and all $t$ in the support of $T$,

$$\mathbb{P}\big(S_t(\tilde{X}_t, Y) \leq u \mid T(\tilde{X}_t) = t\big) \;\leq\; \mathbb{P}\big(S_t(X_t, Y) \leq u \mid T(X_t) = t\big).$$

**Lemma A.2** (A2′ implies $\delta_S \leq \delta_T$). *Let $\delta_S = \sup_u\{\tilde{F}_S(u) - F_S(u)\}_+$ and $\delta_T = \sup_u\{\tilde{F}_T(u) - F_T(u)\}_+$. Under A2′,*

$$\delta_S \;\leq\; \delta_T.$$

*Proof.* Fix $u$ and define $\phi_u(t) := \mathbb{P}(S_t(X_t, Y) \leq u \mid T(X_t) = t)$ and $\tilde{\phi}_u(t) := \mathbb{P}(S_t(\tilde{X}_t, Y) \leq u \mid T(\tilde{X}_t) = t)$. By A2′, $\phi_u$ is nonincreasing and $\tilde{\phi}_u(t) \leq \phi_u(t)$ for all $t$ in the support. Therefore,

$$\tilde{F}_S(u) = \mathbb{E}[\tilde{\phi}_u(T(\tilde{X}_t))] \leq \mathbb{E}[\phi_u(T(\tilde{X}_t))].$$

Thus

$$\tilde{F}_S(u) - F_S(u) \leq \mathbb{E}[\phi_u(T(\tilde{X}_t))] - \mathbb{E}[\phi_u(T(X_t))].$$

Writing the RHS as $\int \phi_u \, d(\tilde{F}_T - F_T)$ and applying integration by parts,

$$\int \phi_u \, d(\tilde{F}_T - F_T) = -\int (\tilde{F}_T - F_T) \, d\phi_u.$$

Since $\phi_u$ is nonincreasing, $-d\phi_u$ is a nonnegative measure with total mass at most 1 (because $\phi_u \in [0,1]$). Hence

$$\tilde{F}_S(u) - F_S(u) \leq \sup_z\{\tilde{F}_T(z) - F_T(z)\}_+ = \delta_T.$$

Taking positive parts and the supremum over $u$ yields $\delta_S \leq \delta_T$. $\qquad\square$

### A.4. A Quantile Lemma

**Lemma A.3** (Quantile inflation under one-sided CDF gap). *Let $q_\gamma$ and $\tilde{q}_\gamma$ denote the $\gamma$-quantiles of $F_S$ and $\tilde{F}_S$. If $\sup_u\{\tilde{F}_S(u) - F_S(u)\}_+ \leq \delta$, then for any $\beta \in (0,1)$,*

$$q_{1-(\beta+\delta)} \;\leq\; \tilde{q}_{1-\beta}.$$

*Proof.* The premise is equivalent to $\tilde{F}_S(u) \leq F_S(u) + \delta$ for all $u$. Let $\tilde{q}_{1-\beta} := \inf\{u : \tilde{F}_S(u) \geq 1 - \beta\}$. Then $1 - \beta \leq \tilde{F}_S(\tilde{q}_{1-\beta}) \leq F_S(\tilde{q}_{1-\beta}) + \delta$, so $F_S(\tilde{q}_{1-\beta}) \geq 1 - (\beta + \delta)$, which implies $\tilde{q}_{1-\beta} \geq q_{1-(\beta+\delta)}$. $\qquad\square$

### A.5. Proof of Theorem 3.2

[Theorem 3.2] Assume A1, A2, and A3. Let $\delta^+$ be defined by (10), $\alpha^\star$ by (11), and $\Gamma_t^{\text{TCC-KS}}$ by (12). Then for any fixed $\eta \in (0,1)$, with probability at least $1 - \eta$ over the unlabeled samples used to compute $\delta^+$,

$$\mathbb{P}\big\{Y \in \Gamma_t^{\text{TCC-KS}}(X_t)\big\} \;\geq\; 1 - \big(\alpha^\star + \delta^+ + \varepsilon\big),$$

where the probability is over both the transported labeled calibration sample $\tilde{\mathcal{C}}_t$ used to form $\hat{q}_{1-\alpha^\star}$ and the test draw $(X_t, Y)$.

*Proof.* Let $\hat{q}_{1-\alpha^\star}$ be the empirical $(1 - \alpha^\star)$ quantile of the transported labeled scores $\{S_t(\tilde{x}_t^{(i)}, y^{(i)})\}_{i=1}^n$, computed from the transported calibration sample $\tilde{\mathcal{C}}_t$. By Assumption A3, standard split-conformal validity on the transported calibration sample implies that for a fresh $(\tilde{X}_t, Y)$ drawn from the transported joint distribution,

$$\mathbb{P}\big(S_t(\tilde{X}_t, Y) \leq \hat{q}_{1-\alpha^\star}\big) \;\geq\; 1 - \alpha^\star, \tag{15}$$

where the probability is over both $\tilde{\mathcal{C}}_t$ (through $\widehat{q}_{1-\alpha^\star}$) and $(\tilde{X}_t, Y)$.

Equivalently, writing $\tilde{F}_S(u) = \mathbb{P}(S_t(\tilde{X}_t, Y) \le u)$ for the transported population CDF and using the tower property,

$$\mathbb{E}_{\tilde{\mathcal{C}}_t}\big[\tilde{F}_S(\widehat{q}_{1-\alpha^\star})\big] \ge 1 - \alpha^\star. \tag{16}$$

Define the one-sided score gap $\delta_S := \sup_u \{\tilde{F}_S(u) - F_S(u)\}_+$. By definition, for all $u$,

$$F_S(u) \ge \tilde{F}_S(u) - \delta_S. \tag{17}$$

Applying (17) at the (random) value $u = \widehat{q}_{1-\alpha^\star}$ and taking expectation over $\tilde{\mathcal{C}}_t$ yields

$$\mathbb{E}_{\tilde{\mathcal{C}}_t}\big[F_S(\widehat{q}_{1-\alpha^\star})\big] \ge \mathbb{E}_{\tilde{\mathcal{C}}_t}\big[\tilde{F}_S(\widehat{q}_{1-\alpha^\star})\big] - \delta_S \ge 1 - \alpha^\star - \delta_S,$$

where the last inequality uses (16).

Since $(X_t, Y)$ is independent of $\tilde{\mathcal{C}}_t$ and has score CDF $F_S$, we have

$$\mathbb{P}\big(S_t(X_t, Y) \le \widehat{q}_{1-\alpha^\star}\big) = \mathbb{E}_{\tilde{\mathcal{C}}_t}\big[F_S(\widehat{q}_{1-\alpha^\star})\big] \ge 1 - \alpha^\star - \delta_S.$$

Equivalently,

$$\mathbb{P}\big\{Y \notin \Gamma_t^{\text{TCC-KS}}(X_t)\big\} = \mathbb{P}\big(S_t(X_t, Y) > \widehat{q}_{1-\alpha^\star}\big) \le \alpha^\star + \delta_S. \tag{18}$$

Assumption A2 states $\delta_S \le \delta_T + \varepsilon$, where $\delta_T = \sup_u \{\tilde{F}_T(u) - F_T(u)\}_+$. By Lemma A.1, with probability at least $1 - \eta$ over the unlabeled samples used to compute $\delta^+$ we have $\delta_T \le \delta^+$. On this event, $\delta_S \le \delta^+ + \varepsilon$, and plugging into (18) yields

$$\mathbb{P}\big\{Y \notin \Gamma_t^{\text{TCC-KS}}(X_t)\big\} \le \alpha^\star + \delta^+ + \varepsilon.$$

Rearranging gives the stated coverage bound. $\qquad\square$

**Corollary (ideal case $\varepsilon = 0$).** If $\varepsilon = 0$ (e.g., when A2$'$ holds so that $\delta_S \le \delta_T$ by Lemma A.2), the theorem reduces to $\mathbb{P}\{Y \in \Gamma_t^{\text{TCC-KS}}(X_t)\} \ge 1 - (\alpha^\star + \delta^+)$, and in particular yields nominal $1 - \alpha$ coverage whenever $\delta^+ \le \alpha$.

### A.6. Why a Surrogate is Needed

Estimating a mismatch bound requires evaluating a statistic on *unlabeled* target inputs. The true-label score $S_t(x, Y)$ is not available without labels, hence we use a label-free surrogate $T(x)$ derived from the predictor. Assumption A2 formalizes a proxy condition under which mismatch in $T(X)$ upper bounds score-relevant mismatch, enabling nontrivial prediction sets. Theorem 3.1 further highlights that, without additional structure of this kind, worst-case valid prediction sets must be essentially vacuous.

### A.7. Proof of Proposition 3.3

We give a self-contained proof for an oracle-weight weighted split-conformal construction, which yields finite-sample validity under covariate shift.

**Weighted Split-Conformal Rule (Oracle-Weight Version).** Let $(\tilde{X}_i, Y_i) \sim \tilde{P}$ for $i = 1, \ldots, n$ denote the transported labeled calibration sample, and let $(X_{n+1}, Y_{n+1}) \sim P$ denote a target test pair. Define scores

$$V_i = S_t(\tilde{X}_i, Y_i) \quad (i \le n), \qquad V_{n+1} = S_t(X_{n+1}, Y_{n+1}),$$

and oracle weights $w(x) = p_{X_t}(x)/p_{\tilde{X}_t}(x)$. For a test input $x$, define $w_{n+1} = w(x)$ and $w_i = w(\tilde{X}_i)$ for $i \le n$. Define the weighted quantile threshold as

$$\widehat{q}_{1-\alpha}^{\text{w}}(x) := \inf\Big\{q \in \mathbb{R} : \sum_{i=1}^n w_i \mathbf{1}\{V_i \le q\} \ge (1-\alpha)\Big(\sum_{i=1}^n w_i + w_{n+1}\Big)\Big\}. \tag{19}$$

Then weighted-TCC predicts

$$\Gamma_t^{\text{wTCC}}(x) = \{y \in \mathcal{Y} : S_t(x, y) \le \widehat{q}_{1-\alpha}^{\text{w}}(x)\}.$$

[Proposition 3.3] Assume (i) covariate shift between transported and target inputs: $\mathcal{L}(Y \mid \tilde{X}_t = x) = \mathcal{L}(Y \mid X_t = x)$ for all $x$, and (ii) $w(x) = p_{X_t}(x)/p_{\tilde{X}_t}(x)$ exists and is used in (19). Then $\Gamma_t^{\mathrm{wTCC}}$ satisfies target-domain marginal coverage

$$\mathbb{P}\{Y_{n+1} \in \Gamma_t^{\mathrm{wTCC}}(X_{n+1})\} \geq 1 - \alpha.$$

*Proof.* Let $(\tilde{X}_i, Y_i)_{i=1}^n$ be i.i.d. from $\tilde{P}$ and let $(X_{n+1}, Y_{n+1})$ be drawn from $P$, independently. Define scores

$$V_i = S_t(\tilde{X}_i, Y_i) \quad (i \leq n), \qquad V_{n+1} = S_t(X_{n+1}, Y_{n+1}),$$

and weights $w_i = w(\tilde{X}_i)$ for $i \leq n$ and $w_{n+1} = w(X_{n+1})$.

Let $\mathcal{Z} = \{(X_1, Y_1), \ldots, (X_{n+1}, Y_{n+1})\}$ denote the *unordered multiset* of the $n + 1$ pairs, where $(X_i, Y_i) = (\tilde{X}_i, Y_i)$ for $i \leq n$ and $(X_{n+1}, Y_{n+1})$ is the target test pair. Let $I \in \{1, \ldots, n+1\}$ denote the (random) index of the element in $\mathcal{Z}$ that was drawn from $P$.

**Step 1 (posterior for the target index).** Under covariate shift, the joint density ratio satisfies

$$\frac{p_P(x, y)}{p_{\tilde{P}}(x, y)} = \frac{p_{X_t}(x)\, p(y \mid x)}{p_{\tilde{X}_t}(x)\, p(y \mid x)} = \frac{p_{X_t}(x)}{p_{\tilde{X}_t}(x)} = w(x).$$

Therefore, by Bayes' rule (conditioning on the multiset $\mathcal{Z}$),

$$\mathbb{P}(I = i \mid \mathcal{Z}) = \frac{w_i}{\sum_{k=1}^{n+1} w_k}. \qquad (*)$$

**Step 2 (compare the split threshold to the full weighted quantile).** Define the *full* weighted $(1 - \alpha)$ quantile

$$q^\star := \inf\left\{ q \in \mathbb{R} : \sum_{k=1}^{n+1} w_k \,\mathbf{1}\{V_k \leq q\} \geq (1 - \alpha) \sum_{k=1}^{n+1} w_k \right\}.$$

Also define the split threshold that would be used if index $i$ were the test point:

$$q_i := \inf\left\{ q \in \mathbb{R} : \sum_{k \neq i} w_k \,\mathbf{1}\{V_k \leq q\} \geq (1 - \alpha) \sum_{k=1}^{n+1} w_k \right\}.$$

(For the realized test input $X_{n+1}$, the algorithm's threshold in (19) is exactly $q_{n+1}$, since $(1 - \alpha)\left(\sum_{k \neq n+1} w_k + w_{n+1}\right) = (1 - \alpha) \sum_{k=1}^{n+1} w_k$.)

Because $\sum_{k \neq i} w_k \mathbf{1}\{V_k \leq q\} \leq \sum_{k=1}^{n+1} w_k \mathbf{1}\{V_k \leq q\}$ for all $q$, the feasibility set defining $q_i$ is a subset of the feasibility set defining $q^\star$, hence

$$q_i \geq q^\star \quad \text{for all } i. \qquad (**)$$

In particular, on the event $\{I = i\}$ we have $\{V_I > q_I\} \subseteq \{V_I > q^\star\}$.

**Step 3 (bound conditional miscoverage).** Conditioning on $\mathcal{Z}$ and using (*) and (**),

$$\mathbb{P}(V_I > q_I \mid \mathcal{Z}) \leq \mathbb{P}(V_I > q^\star \mid \mathcal{Z}) = \sum_{i : V_i > q^\star} \mathbb{P}(I = i \mid \mathcal{Z}) = \sum_{i : V_i > q^\star} \frac{w_i}{\sum_{k=1}^{n+1} w_k}.$$

By the definition of $q^\star$, the total weight of indices with $V_i > q^\star$ is at most an $\alpha$ fraction of the total weight:

$$\sum_{i : V_i > q^\star} w_i \leq \alpha \sum_{k=1}^{n+1} w_k,$$

so $\mathbb{P}(V_I > q_I \mid \mathcal{Z}) \leq \alpha$. Taking expectations yields

$$\mathbb{P}(V_I \leq q_I) \geq 1 - \alpha.$$

Finally, $I$ is (by definition) the index of the target draw from $P$, and $q_I$ is exactly the weighted split threshold used for that target input. Therefore,

$$\mathbb{P}\{Y_{n+1} \in \Gamma_t^{\mathrm{wTCC}}(X_{n+1})\} \geq 1 - \alpha.$$

$\square$

**Practical note.**    weighted-TCC can be sharp when the *residual* shift between $\tilde{X}_t = f(X_s)$ and $X_t$ is close to covariate shift and weights are stable. In practice, weights are estimated and may concentrate (small ESS), which is why we treat weighted-TCC as complementary to the guardrail behavior of TCC-KS.

## B. Implementation Details

This section describes the experimental protocol for TCC-KS, weighted-TCC, and baselines. We report the construction of source and target domains, data splits, model and transport training, calibration procedures, and evaluation metrics.

### B.1. Benchmarks and Domain Construction

**CIFAR-100, Tiny-ImageNet.**    We construct target domains using deterministic image corruptions. We use shot noise, motion blur, brightness, and contrast at severities 1 to 5. For a fixed corruption type and severity, the source inputs are clean images and the target inputs are their corrupted counterparts. Labels are unchanged.

**SAR→RGB.**    We use paired SAR and RGB tiles with four land-cover classes. The source view is SAR and the target view is RGB. No synthetic corruption is applied.

### B.2. Data splits and sample usage

**CIFAR and Tiny-ImageNet.**    For each dataset and each target corruption setting, we partition the clean training set into disjoint subsets. We use 10k labeled examples for source calibration $\mathcal{C}_s$. We use 10k unlabeled target examples as the target pool used by TCC-KS to estimate the surrogate discrepancy. We use 10k paired examples $(x_s, x_t)$ where $x_t$ is the deterministic corruption of $x_s$. We use these pairs to train the transport map and to fit the density ratio model used by weighted-TCC. We use the remaining training examples to train the target classifier $g_t$ on target-domain images.

**SAR→RGB.**    We split paired SAR and RGB tiles into four disjoint subsets of equal size, used for transport training, source calibration, the unlabeled target pool, and the labeled target test set.

**Independence.**    To satisfy Assumption A3, we ensure that the samples used for learning $f$, computing conformal thresholds, estimating the KS discrepancy, and estimating importance weights are disjoint. When disjoint splitting is not feasible, we use cross-fitting.

### B.3. Target Predictor and Nonconformity Score

**Target classifier.**    For CIFAR and Tiny-ImageNet, the target predictor $g_t$ is a ResNet-18 trained on target-domain images using Adam with learning rate $10^{-3}$, batch size 128, and two epochs. For SAR→RGB, $g_t$ is a ResNet-18 trained on RGB tiles for a four-class task (agri, barrenland, grassland, and urban). When a method requires evaluating $g_t$ on SAR inputs (e.g., the no-transport baseline), we first apply a fixed, deterministic channel adapter to match the 3-channel input format expected by the RGB-trained network.

**Score.**    All prediction sets use the least ambiguous class (LAC) score

$$S_t(x, y) \;=\; 1 - \hat{p}_t(y \mid x), \qquad \hat{p}_t(\cdot \mid x) = \mathrm{softmax}(g_t(x)).$$

We evaluate $\alpha \in \{0.1, 0.2, 0.3, 0.4\}$ when sweeping nominal levels. We compute split-conformal thresholds using exact order statistics with the standard right-continuous quantile convention.

### B.4. Transport map learning

**CIFAR and Tiny-ImageNet.**    We learn a transport map $f : \mathcal{X}_s \to \mathcal{X}_t$ on paired clean to corrupted images using a convolutional encoder-decoder. The encoder uses channel widths 3, 32, 64, and 128 with downsampling by pooling. The decoder uses channel widths 128, 64, and 32 with upsampling by transposed convolutions and a sigmoid output. We train with an $\ell_1$ reconstruction objective $\|f(x_s) - x_t\|_1$.

We also evaluate a predictive alignment variant that adds a label-free KL term on paired inputs

$$\mathcal{L}_{\mathrm{trans}} \;=\; \|f(x_s) - x_t\|_1 + \lambda \, \mathrm{KL}\big(\hat{p}_t(\cdot \mid f(x_s)) \, \| \, \hat{p}_t(\cdot \mid x_t)\big),$$

with $\lambda = 0.5$. We evaluate multiple transport checkpoints at epochs 2, 5, 10, 20, 50, 80, and 150.

**SAR→RGB.** We learn $f$ using a pix2pix-style U-Net generator trained on paired SAR and RGB tiles with an $\ell_1$ objective and the optional predictive KL term above. We use Adam with learning rate $10^{-3}$ and betas $(0.5, 0.999)$.

### B.5. TCC-KS Implementation

**Surrogate.** We use the uncertainty surrogate

$$T(x) \;=\; 1 - \max_{y \in \mathcal{Y}} \hat{p}_t(y \mid x),$$

which is computable without labels.

**One-sided KS Discrepancy.** Using unlabeled samples from the target pool and transported samples $\tilde{x}_t = f(x_s)$, we compute empirical CDFs $\widehat{F}_T$ and $\widehat{\tilde{F}}_T$ and estimate the one-sided gap in the target-harder direction

$$\hat{\delta} \;=\; \sup_{u \in \mathbb{R}} \big\{ \widehat{\tilde{F}}_T(u) - \widehat{F}_T(u) \big\}_+.$$

We form a high-probability upper bound $\delta^+$ using the two-sample DKW inflation stated in the main text with confidence parameter $\eta = 0.1$.

**Adjusted Level and Calibration.** We set
$$\alpha^\star \;=\; \max(0, \alpha - \delta^+),$$

and compute the transported split-conformal threshold at level $\alpha^\star$ using transported labeled calibration scores $\{S_t(f(x_s^{(i)}), y^{(i)})\}_{i=1}^n$.

### B.6. Weighted-TCC Implementation

weighted-TCC performs importance weighting after transport, treating $\tilde{X}_t = f(X_s)$ as the labeled calibration distribution and $X_t$ as deployment.

**Density Ratio Estimation.** We estimate density ratios via a domain classifier that distinguishes real target inputs $x_t$ from transported inputs $\tilde{x}_t = f(x_s)$ (both in the target input format) using features extracted by the target predictor. Concretely, we use the target-model logits $g_t(x)$ as features and fit a logistic-regression classifier on up to 5k samples from each domain for 20 epochs with learning rate $10^{-3}$. This yields an estimate of the density ratio in the induced logit space (i.e., shift as reflected by the predictor's outputs), which is more stable than ratio estimation in raw pixel space.

**Weights.** Let $\hat{p}_{\mathrm{dom}}(x)$ denote the classifier probability that $x$ is a real target input. We define weights via odds

$$w(x) \;=\; \frac{\hat{p}_{\mathrm{dom}}(x)}{1 - \hat{p}_{\mathrm{dom}}(x)},$$

and clip weights at 5.

**Weighted Conformal Threshold.** We compute a weighted conformal threshold on transported labeled calibration scores using weights $w(\tilde{X}_i)$ and apply the resulting prediction sets on target test inputs.

### B.7. Baselines

We compare against the following baselines. Oracle CP calibrates using labeled target counterparts of the source calibration examples and is reported as a reference. Target-only CP calibrates using labeled target calibration data drawn from the target pool. Transport-only CP calibrates on transported labeled scores without KS correction and without weighting. Weighted CP without transport applies importance-weighted conformal calibration with the identity map in place of $f$.

## B.8. Evaluation Metrics and Diagnostics

All metrics are computed on a held-out labeled target test set. Labels are used only for evaluation.

**Empirical Marginal Coverage.** Given target test examples $\{(X_{t,i}, Y_i)\}_{i=1}^{n_\text{test}}$ and prediction sets $\Gamma(\cdot)$, we report

$$\widehat{\text{cov}} \;=\; \frac{1}{n_\text{test}} \sum_{i=1}^{n_\text{test}} \mathbf{1}\{Y_i \in \Gamma(X_{t,i})\}.$$

**Average Prediction Set Size.** We report

$$\widehat{\text{size}} \;=\; \frac{1}{n_\text{test}} \sum_{i=1}^{n_\text{test}} |\Gamma(X_{t,i})|.$$

**KS-Aligned Diagnostic for TCC-KS.** We report the estimated upper bound $\delta^+$ and the resulting worst-case miscoverage bound

$$\alpha_\text{bnd} \;=\; \alpha^\star + \delta^+, \qquad \alpha^\star = \max(0, \alpha - \delta^+).$$

We emphasize that $\alpha_\text{bnd}$ is a conservative certificate and is not expected to match empirical miscoverage under strong shift.

**Weight Stability for Weighted-TCC.** Given weights $\{w_j\}_{j=1}^N$ computed on the unlabeled sample used for weight estimation, we report the effective sample size

$$\text{ESS} \;=\; \frac{\left(\sum_{j=1}^N w_j\right)^2}{\sum_{j=1}^N w_j^2},$$

and its normalized form

$$\text{ESS\%} \;=\; 100 \cdot \frac{\text{ESS}}{N}.$$

**Surrogate–Score Alignment Audit.** To probe Assumptions A2 and A2′, we compute diagnostics on labeled target test data and on transported labeled test data. We report Spearman rank correlation between surrogate and true-label score,

$$\rho(T, S) \;=\; \text{Corr}\big(\text{rank}(T(X)), \text{rank}(S_t(X, Y))\big).$$

We also compute bin-wise monotonicity diagnostics using $B = 10$ quantile bins of the surrogate. Let $\mathcal{I}_b$ be the indices in bin $b$, ordered by increasing surrogate level. Define the bin statistic

$$\mu_b \;=\; \frac{1}{|\mathcal{I}_b|} \sum_{i \in \mathcal{I}_b} S_t(X_i, Y_i), \qquad q_{0.9,b} \;=\; \inf\left\{ u : \frac{1}{|\mathcal{I}_b|} \sum_{i \in \mathcal{I}_b} \mathbf{1}\{S_t(X_i, Y_i) \le u\} \ge 0.9 \right\}.$$

We report violation rates for mean and upper-tail monotonicity as

$$\text{Viol}_\text{mean} \;=\; \frac{1}{B-1} \sum_{b=1}^{B-1} \mathbf{1}\{\mu_{b+1} < \mu_b\}, \qquad \text{Viol}_{q90} \;=\; \frac{1}{B-1} \sum_{b=1}^{B-1} \mathbf{1}\{q_{0.9,b+1} < q_{0.9,b}\}.$$

**Averaging Across Target Conditions.** For corrupted-image benchmarks we often report averages across corruption types and severities. If there are $K$ target conditions and a metric $M_k$ computed per condition, we report the macro-average

$$\overline{M} \;=\; \frac{1}{K} \sum_{k=1}^K M_k.$$

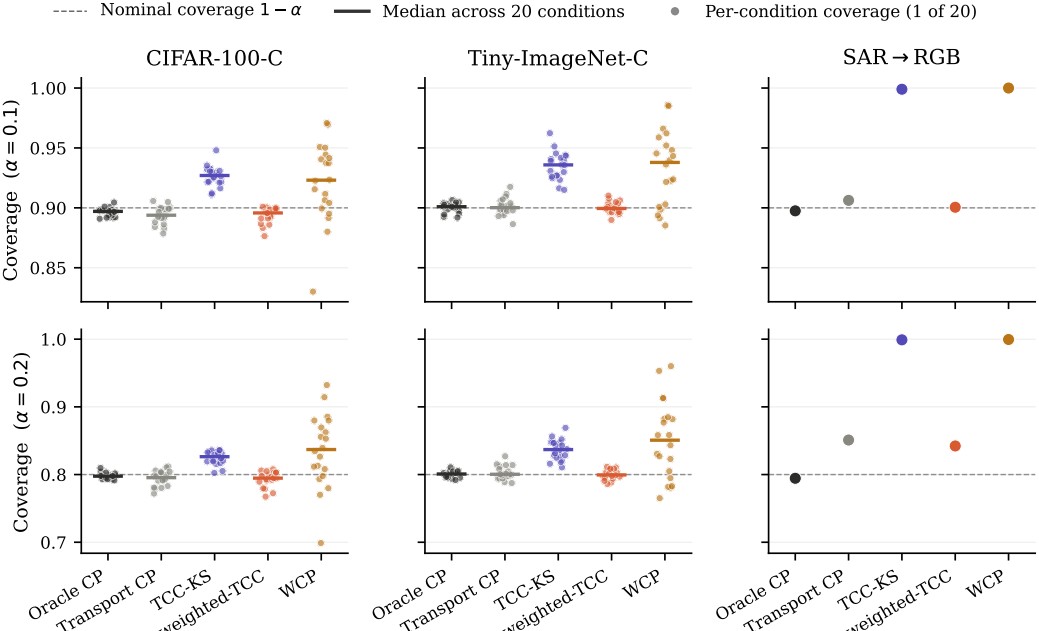

*Figure 3.* Per-condition coverage distributions for the settings summarized in Table 1. For CIFAR-100-C and Tiny-ImageNet-C, each dot corresponds to one of the 20 target conditions (4 corruptions $\times$ 5 severities; epoch 150, $\ell_1$+KL transport), and the horizontal bar marks the median across conditions. SAR$\rightarrow$RGB has a single evaluation setting and is shown as one point per method. The dashed line marks nominal coverage $1 - \alpha$. The distributions show that Transport CP and weighted-TCC closely track Oracle CP across conditions, while TCC-KS consistently shifts coverage upward as a $\delta^+$-driven validity guardrail. WCP exhibits the largest per-condition variability, including individual under-coverage cases.

## C. Detailed Results for Experiment 1

To expose better the per-corruption variability in Table 1 and Table 2, Figure 3 shows the per-condition coverage distributions, and Figure 4 shows the corresponding average set-size distributions. Each point corresponds to one target condition, and horizontal bars denote medians across conditions.

In addition to Figures 3 and 4, we report results for each corruption for severities that are mild (severity=3) and high (severity=5) for $\alpha \in \{0.1, 0.2, 0.3, 0.4\}$ and using the $\ell_1 + \text{KL}$ transport loss at epoch 150.

|  | Severity 3 | | | | | | | | | | | | Severity 5 | | | | | | | | | | | |
| --- | --- | --- | --- | --- | --- | --- | --- | --- | --- | --- | --- | --- | --- | --- | --- | --- | --- | --- | --- | --- | --- | --- | --- | --- |
|  | $\alpha_{\text{nom}}{=}0.10$ | | | $\alpha_{\text{nom}}{=}0.20$ | | | $\alpha_{\text{nom}}{=}0.30$ | | | $\alpha_{\text{nom}}{=}0.40$ | | | $\alpha_{\text{nom}}{=}0.10$ | | | $\alpha_{\text{nom}}{=}0.20$ | | | $\alpha_{\text{nom}}{=}0.30$ | | | $\alpha_{\text{nom}}{=}0.40$ | | |
| Method | Cov | Size | $\delta^+$ | Cov | Size | $\delta^+$ | Cov | Size | $\delta^+$ | Cov | Size | $\delta^+$ | Cov | Size | $\delta^+$ | Cov | Size | $\delta^+$ | Cov | Size | $\delta^+$ | Cov | Size | $\delta^+$ |
| Oracle CP | 0.90 | 51.21 | 0.03 | 0.81 | 33.54 | 0.03 | 0.70 | 22.70 | 0.03 | 0.61 | 15.78 | 0.03 | 0.89 | 39.06 | 0.01 | 0.78 | 25.60 | 0.01 | 0.68 | 17.44 | 0.01 | 0.58 | 11.93 | 0.01 |
| Transport only | 0.89 | 47.66 | 0.03 | 0.80 | 31.03 | 0.03 | 0.69 | 20.73 | 0.03 | 0.59 | 14.14 | 0.03 | 0.89 | 38.65 | 0.01 | 0.78 | 25.30 | 0.01 | 0.67 | 17.13 | 0.01 | 0.58 | 11.73 | 0.01 |
| WCP-no-transport | 0.83 | 36.69 | 0.03 | 0.70 | 22.79 | 0.03 | 0.58 | 14.27 | 0.03 | 0.47 | 8.87 | 0.03 | 0.92 | 44.87 | 0.01 | 0.85 | 32.03 | 0.01 | 0.76 | 23.30 | 0.01 | 0.66 | 16.41 | 0.01 |
| weighted-TCC | 0.90 | 49.26 | 0.03 | 0.80 | 32.10 | 0.03 | 0.69 | 21.45 | 0.03 | 0.59 | 14.67 | 0.03 | 0.88 | 36.51 | 0.01 | 0.77 | 23.66 | 0.01 | 0.67 | 16.08 | 0.01 | 0.57 | 11.02 | 0.01 |
| TCC-KS | 0.93 | 56.74 | 0.03 | 0.84 | 37.76 | 0.03 | 0.74 | 25.97 | 0.03 | 0.64 | 18.17 | 0.03 | 0.93 | 46.39 | 0.01 | 0.83 | 31.15 | 0.01 | 0.74 | 21.44 | 0.01 | 0.64 | 14.99 | 0.01 |

*Table 6.* Coverage, average prediction set size, and empirical $\delta^+$ on CIFAR-100-C (brightness) using the $\ell_1 + \text{KL}$ transport objective at epoch 150.

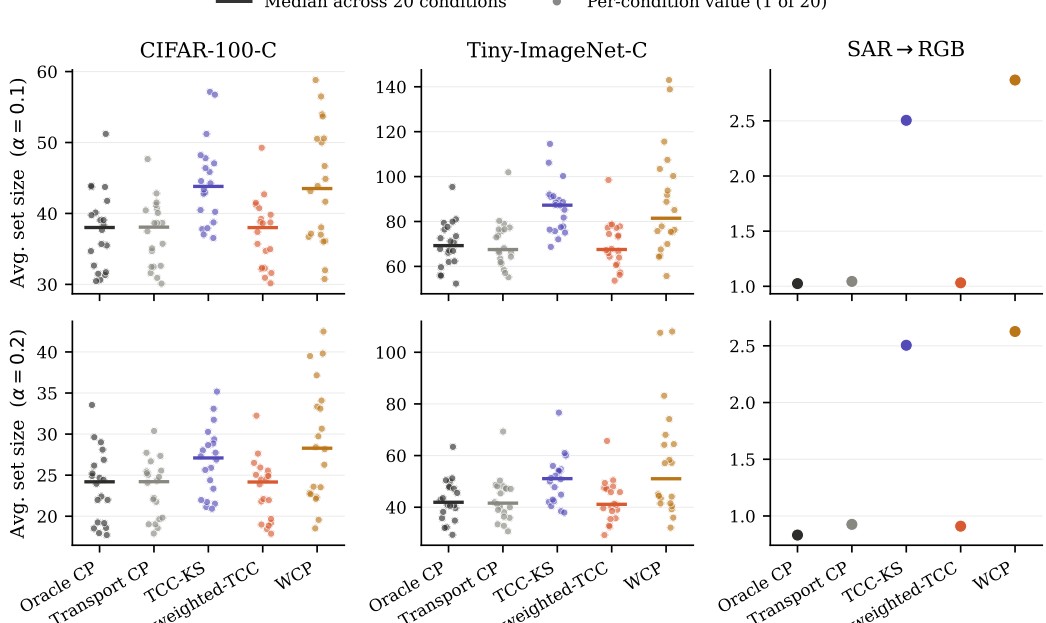

*Figure 4.* Per-condition average prediction-set-size distributions for the settings summarized in Table 1. Conventions follow Figure 3. The y-axes are scaled separately across datasets because absolute set sizes differ substantially (CIFAR-100-C: roughly 30–60; Tiny-ImageNet-C: roughly 60–140; SAR→RGB: roughly 1–3). Transport CP and weighted-TCC remain close to Oracle CP in both median and spread, whereas TCC-KS incurs a moderate efficiency cost for its $\delta^+$-driven safety margin. WCP shows the widest set-size spread, matching its larger coverage variability in Figure 3.

| | Severity 3 | | | | | | | | | | | | Severity 5 | | | | | | | | | | | |
| | $\alpha_{\mathrm{nom}}$=0.10 | | | $\alpha_{\mathrm{nom}}$=0.20 | | | $\alpha_{\mathrm{nom}}$=0.30 | | | $\alpha_{\mathrm{nom}}$=0.40 | | | $\alpha_{\mathrm{nom}}$=0.10 | | | $\alpha_{\mathrm{nom}}$=0.20 | | | $\alpha_{\mathrm{nom}}$=0.30 | | | $\alpha_{\mathrm{nom}}$=0.40 | | |
| Method | Cov | Size | $\delta^+$ | Cov | Size | $\delta^+$ | Cov | Size | $\delta^+$ | Cov | Size | $\delta^+$ | Cov | Size | $\delta^+$ | Cov | Size | $\delta^+$ | Cov | Size | $\delta^+$ | Cov | Size | $\delta^+$ |
|---|---|---|---|---|---|---|---|---|---|---|---|---|---|---|---|---|---|---|---|---|---|---|---|---|
| Oracle CP | 0.89 | 27.51 | 0.02 | 0.79 | 17.94 | 0.02 | 0.69 | 11.63 | 0.02 | 0.59 | 7.57 | 0.02 | 0.89 | 24.61 | 0.02 | 0.79 | 15.93 | 0.02 | 0.68 | 10.39 | 0.02 | 0.58 | 6.77 | 0.02 |
| Transport only | 0.88 | 25.92 | 0.02 | 0.81 | 19.64 | 0.02 | 0.72 | 14.54 | 0.02 | 0.62 | 10.55 | 0.02 | 0.89 | 25.07 | 0.02 | 0.82 | 18.15 | 0.02 | 0.72 | 13.05 | 0.02 | 0.62 | 9.25 | 0.02 |
| WCP-no-transport | 0.76 | 18.72 | 0.02 | 0.88 | 28.16 | 0.02 | 0.82 | 21.76 | 0.02 | 0.74 | 16.61 | 0.02 | 0.84 | 26.47 | 0.02 | 0.79 | 19.33 | 0.02 | 0.73 | 14.06 | 0.02 | 0.64 | 10.08 | 0.02 |
| weighted-TCC | 0.88 | 25.00 | 0.02 | 0.80 | 18.99 | 0.02 | 0.71 | 13.98 | 0.02 | 0.61 | 10.03 | 0.02 | 0.89 | 24.46 | 0.02 | 0.80 | 17.57 | 0.02 | 0.71 | 12.62 | 0.02 | 0.61 | 8.91 | 0.02 |
| TCC-KS | 0.92 | 30.89 | 0.02 | 0.83 | 21.97 | 0.02 | 0.74 | 15.96 | 0.02 | 0.64 | 11.40 | 0.02 | 0.93 | 31.03 | 0.02 | 0.84 | 22.28 | 0.02 | 0.75 | 16.06 | 0.02 | 0.65 | 11.43 | 0.02 |

*Table 7.* Coverage, average prediction set size, and empirical $\delta^+$ on CIFAR-100-C (contrast) using the $\ell_1 + \mathrm{KL}$ transport objective at epoch 150.

| | Severity 3 | | | | | | | | | | | | Severity 5 | | | | | | | | | | | |
| | $\alpha_{\mathrm{nom}}$=0.10 | | | $\alpha_{\mathrm{nom}}$=0.20 | | | $\alpha_{\mathrm{nom}}$=0.30 | | | $\alpha_{\mathrm{nom}}$=0.40 | | | $\alpha_{\mathrm{nom}}$=0.10 | | | $\alpha_{\mathrm{nom}}$=0.20 | | | $\alpha_{\mathrm{nom}}$=0.30 | | | $\alpha_{\mathrm{nom}}$=0.40 | | |
| Method | Cov | Size | $\delta^+$ | Cov | Size | $\delta^+$ | Cov | Size | $\delta^+$ | Cov | Size | $\delta^+$ | Cov | Size | $\delta^+$ | Cov | Size | $\delta^+$ | Cov | Size | $\delta^+$ | Cov | Size | $\delta^+$ |
|---|---|---|---|---|---|---|---|---|---|---|---|---|---|---|---|---|---|---|---|---|---|---|---|---|
| Oracle CP | 0.90 | 33.10 | 0.02 | 0.80 | 21.66 | 0.02 | 0.70 | 14.40 | 0.02 | 0.60 | 9.57 | 0.02 | 0.90 | 35.03 | 0.03 | 0.80 | 22.90 | 0.03 | 0.70 | 15.12 | 0.03 | 0.60 | 10.03 | 0.03 |
| Transport only | 0.89 | 31.55 | 0.02 | 0.79 | 20.89 | 0.02 | 0.69 | 14.11 | 0.02 | 0.59 | 9.63 | 0.02 | 0.89 | 34.40 | 0.03 | 0.79 | 22.59 | 0.03 | 0.69 | 15.21 | 0.03 | 0.59 | 10.24 | 0.03 |
| WCP-no-transport | 0.91 | 35.11 | 0.02 | 0.82 | 23.24 | 0.02 | 0.71 | 15.64 | 0.02 | 0.60 | 10.63 | 0.02 | 0.93 | 40.08 | 0.03 | 0.86 | 28.13 | 0.03 | 0.78 | 20.25 | 0.03 | 0.68 | 14.51 | 0.03 |
| weighted-TCC | 0.89 | 31.71 | 0.02 | 0.79 | 20.98 | 0.02 | 0.69 | 14.18 | 0.02 | 0.59 | 9.68 | 0.02 | 0.90 | 34.90 | 0.03 | 0.80 | 22.85 | 0.03 | 0.70 | 15.38 | 0.03 | 0.60 | 10.35 | 0.03 |
| TCC-KS | 0.93 | 38.08 | 0.02 | 0.84 | 25.59 | 0.02 | 0.74 | 17.49 | 0.02 | 0.64 | 12.13 | 0.02 | 0.95 | 45.43 | 0.03 | 0.88 | 31.43 | 0.03 | 0.80 | 22.26 | 0.03 | 0.71 | 15.74 | 0.03 |

*Table 8.* Coverage, average prediction set size, and empirical $\delta^+$ on CIFAR-100-C (motion-blur) using the $\ell_1 + \mathrm{KL}$ transport objective at epoch 150.

| | Severity 3 | | | | | | | | | | | | Severity 5 | | | | | | | | | | | |
|---|---|---|---|---|---|---|---|---|---|---|---|---|---|---|---|---|---|---|---|---|---|---|---|---|---|
| | $\alpha_{\text{nom}}$=0.10 | | | $\alpha_{\text{nom}}$=0.20 | | | $\alpha_{\text{nom}}$=0.30 | | | $\alpha_{\text{nom}}$=0.40 | | | $\alpha_{\text{nom}}$=0.10 | | | $\alpha_{\text{nom}}$=0.20 | | | $\alpha_{\text{nom}}$=0.30 | | | $\alpha_{\text{nom}}$=0.40 | | |
| Method | Cov | Size | $\delta^+$ | Cov | Size | $\delta^+$ | Cov | Size | $\delta^+$ | Cov | Size | $\delta^+$ | Cov | Size | $\delta^+$ | Cov | Size | $\delta^+$ | Cov | Size | $\delta^+$ | Cov | Size | $\delta^+$ |
| Oracle CP | 0.90 | 46.07 | 0.03 | 0.80 | 30.17 | 0.03 | 0.70 | 20.02 | 0.03 | 0.60 | 13.50 | 0.03 | 0.90 | 54.60 | 0.04 | 0.80 | 36.55 | 0.04 | 0.70 | 24.76 | 0.04 | 0.60 | 16.88 | 0.04 |
| Transport only | 0.89 | 44.16 | 0.03 | 0.79 | 28.83 | 0.03 | 0.69 | 19.02 | 0.03 | 0.59 | 12.77 | 0.03 | 0.90 | 54.26 | 0.04 | 0.80 | 36.21 | 0.04 | 0.70 | 24.52 | 0.04 | 0.60 | 16.63 | 0.04 |
| WCP-no-transport | 0.91 | 50.20 | 0.03 | 0.83 | 34.35 | 0.03 | 0.74 | 24.03 | 0.03 | 0.64 | 17.01 | 0.03 | 0.94 | 64.10 | 0.04 | 0.88 | 45.52 | 0.04 | 0.81 | 32.27 | 0.04 | 0.73 | 23.18 | 0.04 |
| weighted-TCC | 0.89 | 44.83 | 0.03 | 0.79 | 29.26 | 0.03 | 0.69 | 19.30 | 0.03 | 0.59 | 12.96 | 0.03 | 0.90 | 54.75 | 0.04 | 0.80 | 36.44 | 0.04 | 0.70 | 24.66 | 0.04 | 0.60 | 16.74 | 0.04 |
| TCC-KS | 0.93 | 54.74 | 0.03 | 0.85 | 37.36 | 0.03 | 0.77 | 26.09 | 0.03 | 0.68 | 18.55 | 0.03 | 0.96 | 70.43 | 0.04 | 0.90 | 50.37 | 0.04 | 0.84 | 36.45 | 0.04 | 0.77 | 26.59 | 0.04 |

*Table 9.* Coverage, average prediction set size, and empirical $\delta^+$ on CIFAR-100-C (shot-noise) using the $\ell_1 + \text{KL}$ transport objective at epoch 150.

| | Severity 3 | | | | | | | | | | | | Severity 5 | | | | | | | | | | | |
|---|---|---|---|---|---|---|---|---|---|---|---|---|---|---|---|---|---|---|---|---|---|---|---|---|---|
| | $\alpha_{\text{nom}}$=0.10 | | | $\alpha_{\text{nom}}$=0.20 | | | $\alpha_{\text{nom}}$=0.30 | | | $\alpha_{\text{nom}}$=0.40 | | | $\alpha_{\text{nom}}$=0.10 | | | $\alpha_{\text{nom}}$=0.20 | | | $\alpha_{\text{nom}}$=0.30 | | | $\alpha_{\text{nom}}$=0.40 | | |
| Method | Cov | Size | $\delta^+$ | Cov | Size | $\delta^+$ | Cov | Size | $\delta^+$ | Cov | Size | $\delta^+$ | Cov | Size | $\delta^+$ | Cov | Size | $\delta^+$ | Cov | Size | $\delta^+$ | Cov | Size | $\delta^+$ |
| Oracle CP | 0.90 | 96.35 | 0.02 | 0.80 | 73.55 | 0.02 | 0.70 | 55.59 | 0.02 | 0.60 | 41.24 | 0.02 | 0.90 | 91.45 | 0.01 | 0.80 | 69.59 | 0.01 | 0.70 | 52.32 | 0.01 | 0.60 | 38.43 | 0.01 |
| Transport only | 0.89 | 94.77 | 0.02 | 0.79 | 72.27 | 0.02 | 0.69 | 54.65 | 0.02 | 0.59 | 40.49 | 0.02 | 0.89 | 91.17 | 0.01 | 0.79 | 69.37 | 0.01 | 0.69 | 52.21 | 0.01 | 0.59 | 38.32 | 0.01 |
| WCP-no-transport | 0.92 | 103.23 | 0.02 | 0.85 | 82.93 | 0.02 | 0.77 | 67.34 | 0.02 | 0.69 | 55.05 | 0.02 | 0.93 | 98.88 | 0.01 | 0.87 | 78.69 | 0.01 | 0.80 | 63.77 | 0.01 | 0.73 | 51.97 | 0.01 |
| weighted-TCC | 0.89 | 95.09 | 0.02 | 0.79 | 72.35 | 0.02 | 0.69 | 54.66 | 0.02 | 0.59 | 40.47 | 0.02 | 0.89 | 91.32 | 0.01 | 0.79 | 69.46 | 0.01 | 0.69 | 52.29 | 0.01 | 0.59 | 38.37 | 0.01 |
| TCC-KS | 0.93 | 107.49 | 0.02 | 0.87 | 86.28 | 0.02 | 0.80 | 70.62 | 0.02 | 0.72 | 58.12 | 0.02 | 0.94 | 102.89 | 0.01 | 0.89 | 82.12 | 0.01 | 0.83 | 67.01 | 0.01 | 0.77 | 54.89 | 0.01 |

*Table 10.* Coverage, average prediction set size, and empirical $\delta^+$ on Tiny-ImageNet-C (brightness) using the $\ell_1 + \text{KL}$ transport objective at epoch 150.

| | Severity 3 | | | | | | | | | | | | Severity 5 | | | | | | | | | | | |
|---|---|---|---|---|---|---|---|---|---|---|---|---|---|---|---|---|---|---|---|---|---|---|---|---|---|
| | $\alpha_{\text{nom}}$=0.10 | | | $\alpha_{\text{nom}}$=0.20 | | | $\alpha_{\text{nom}}$=0.30 | | | $\alpha_{\text{nom}}$=0.40 | | | $\alpha_{\text{nom}}$=0.10 | | | $\alpha_{\text{nom}}$=0.20 | | | $\alpha_{\text{nom}}$=0.30 | | | $\alpha_{\text{nom}}$=0.40 | | |
| Method | Cov | Size | $\delta^+$ | Cov | Size | $\delta^+$ | Cov | Size | $\delta^+$ | Cov | Size | $\delta^+$ | Cov | Size | $\delta^+$ | Cov | Size | $\delta^+$ | Cov | Size | $\delta^+$ | Cov | Size | $\delta^+$ |
| Oracle CP | 0.90 | 95.04 | 0.03 | 0.80 | 72.45 | 0.03 | 0.70 | 54.88 | 0.03 | 0.60 | 40.98 | 0.03 | 0.90 | 97.29 | 0.02 | 0.80 | 74.58 | 0.02 | 0.70 | 56.78 | 0.02 | 0.60 | 42.58 | 0.02 |
| Transport only | 0.89 | 93.66 | 0.03 | 0.79 | 71.24 | 0.03 | 0.69 | 53.78 | 0.03 | 0.59 | 40.04 | 0.03 | 0.89 | 96.15 | 0.02 | 0.79 | 73.40 | 0.02 | 0.69 | 55.62 | 0.02 | 0.59 | 41.54 | 0.02 |
| WCP-no-transport | 0.93 | 101.96 | 0.03 | 0.86 | 81.72 | 0.03 | 0.78 | 66.25 | 0.03 | 0.70 | 54.08 | 0.03 | 0.94 | 105.14 | 0.02 | 0.89 | 85.16 | 0.02 | 0.83 | 69.75 | 0.02 | 0.77 | 57.40 | 0.02 |
| weighted-TCC | 0.89 | 94.01 | 0.03 | 0.79 | 71.31 | 0.03 | 0.69 | 53.77 | 0.03 | 0.59 | 40.01 | 0.03 | 0.89 | 96.48 | 0.02 | 0.79 | 73.48 | 0.02 | 0.69 | 55.63 | 0.02 | 0.59 | 41.54 | 0.02 |
| TCC-KS | 0.94 | 105.99 | 0.03 | 0.88 | 84.87 | 0.03 | 0.81 | 69.30 | 0.03 | 0.74 | 57.05 | 0.03 | 0.95 | 109.58 | 0.02 | 0.90 | 89.17 | 0.02 | 0.85 | 74.03 | 0.02 | 0.79 | 62.00 | 0.02 |

*Table 11.* Coverage, average prediction set size, and empirical $\delta^+$ on Tiny-ImageNet-C (contrast) using the $\ell_1 + \text{KL}$ transport objective at epoch 150.

| | Severity 3 | | | | | | | | | | | | Severity 5 | | | | | | | | | | | |
|---|---|---|---|---|---|---|---|---|---|---|---|---|---|---|---|---|---|---|---|---|---|---|---|---|---|
| | $\alpha_{\text{nom}}$=0.10 | | | $\alpha_{\text{nom}}$=0.20 | | | $\alpha_{\text{nom}}$=0.30 | | | $\alpha_{\text{nom}}$=0.40 | | | $\alpha_{\text{nom}}$=0.10 | | | $\alpha_{\text{nom}}$=0.20 | | | $\alpha_{\text{nom}}$=0.30 | | | $\alpha_{\text{nom}}$=0.40 | | |
| Method | Cov | Size | $\delta^+$ | Cov | Size | $\delta^+$ | Cov | Size | $\delta^+$ | Cov | Size | $\delta^+$ | Cov | Size | $\delta^+$ | Cov | Size | $\delta^+$ | Cov | Size | $\delta^+$ | Cov | Size | $\delta^+$ |
| Oracle CP | 0.90 | 95.48 | 0.03 | 0.80 | 72.83 | 0.03 | 0.70 | 55.18 | 0.03 | 0.60 | 41.23 | 0.03 | 0.90 | 97.80 | 0.04 | 0.80 | 74.95 | 0.04 | 0.70 | 57.04 | 0.04 | 0.60 | 42.80 | 0.04 |
| Transport only | 0.89 | 94.37 | 0.03 | 0.79 | 71.85 | 0.03 | 0.69 | 54.22 | 0.03 | 0.59 | 40.28 | 0.03 | 0.89 | 97.47 | 0.04 | 0.79 | 74.56 | 0.04 | 0.69 | 56.46 | 0.04 | 0.59 | 42.05 | 0.04 |
| WCP-no-transport | 0.93 | 102.84 | 0.03 | 0.86 | 82.65 | 0.03 | 0.78 | 66.95 | 0.03 | 0.70 | 54.65 | 0.03 | 0.95 | 110.18 | 0.04 | 0.91 | 91.06 | 0.04 | 0.87 | 76.87 | 0.04 | 0.83 | 65.66 | 0.04 |
| weighted-TCC | 0.89 | 94.72 | 0.03 | 0.79 | 71.92 | 0.03 | 0.69 | 54.23 | 0.03 | 0.59 | 40.28 | 0.03 | 0.89 | 97.74 | 0.04 | 0.79 | 74.61 | 0.04 | 0.69 | 56.44 | 0.04 | 0.59 | 42.03 | 0.04 |
| TCC-KS | 0.94 | 108.25 | 0.03 | 0.88 | 87.09 | 0.03 | 0.82 | 71.50 | 0.03 | 0.75 | 59.15 | 0.03 | 0.96 | 117.64 | 0.04 | 0.93 | 99.20 | 0.04 | 0.90 | 85.73 | 0.04 | 0.86 | 75.27 | 0.04 |

*Table 12.* Coverage, average prediction set size, and empirical $\delta^+$ on Tiny-ImageNet-C (motion-blur) using the $\ell_1 + \text{KL}$ transport objective at epoch 150.

| | Severity 3 | | | | | | | | | | | | Severity 5 | | | | | | | | | | | |
|---|---|---|---|---|---|---|---|---|---|---|---|---|---|---|---|---|---|---|---|---|---|---|---|---|---|
| | $\alpha_{nom}=0.10$ | | | $\alpha_{nom}=0.20$ | | | $\alpha_{nom}=0.30$ | | | $\alpha_{nom}=0.40$ | | | $\alpha_{nom}=0.10$ | | | $\alpha_{nom}=0.20$ | | | $\alpha_{nom}=0.30$ | | | $\alpha_{nom}=0.40$ | | |
| Method | Cov | Size | $\delta^+$ | Cov | Size | $\delta^+$ | Cov | Size | $\delta^+$ | Cov | Size | $\delta^+$ | Cov | Size | $\delta^+$ | Cov | Size | $\delta^+$ | Cov | Size | $\delta^+$ | Cov | Size | $\delta^+$ |
| Oracle CP | 0.90 | 95.78 | 0.03 | 0.80 | 73.07 | 0.03 | 0.70 | 55.46 | 0.03 | 0.60 | 41.46 | 0.03 | 0.90 | 98.23 | 0.04 | 0.80 | 75.16 | 0.04 | 0.70 | 56.97 | 0.04 | 0.60 | 42.58 | 0.04 |
| Transport only | 0.89 | 94.96 | 0.03 | 0.79 | 72.39 | 0.03 | 0.69 | 54.80 | 0.03 | 0.59 | 40.85 | 0.03 | 0.89 | 97.84 | 0.04 | 0.79 | 74.65 | 0.04 | 0.69 | 56.42 | 0.04 | 0.59 | 42.02 | 0.04 |
| WCP-no-transport | 0.93 | 103.76 | 0.03 | 0.87 | 83.57 | 0.03 | 0.79 | 67.88 | 0.03 | 0.71 | 55.42 | 0.03 | 0.95 | 111.21 | 0.04 | 0.92 | 92.13 | 0.04 | 0.88 | 77.88 | 0.04 | 0.84 | 66.56 | 0.04 |
| weighted-TCC | 0.89 | 95.31 | 0.03 | 0.79 | 72.42 | 0.03 | 0.69 | 54.80 | 0.03 | 0.59 | 40.83 | 0.03 | 0.89 | 98.11 | 0.04 | 0.79 | 74.70 | 0.04 | 0.69 | 56.40 | 0.04 | 0.59 | 42.01 | 0.04 |
| TCC-KS | 0.94 | 109.27 | 0.03 | 0.88 | 88.12 | 0.03 | 0.82 | 72.55 | 0.03 | 0.75 | 60.09 | 0.03 | 0.96 | 118.69 | 0.04 | 0.93 | 100.13 | 0.04 | 0.90 | 86.41 | 0.04 | 0.86 | 75.77 | 0.04 |

*Table 13.* Coverage, average prediction set size, and empirical $\delta^+$ on Tiny-ImageNet-C (shot-noise) using the $\ell_1 + \mathrm{KL}$ transport objective at epoch 150.

## D. Detailed Results for Experiment 2

We report results for different epochs and losses for each data set for $\alpha \in \{0.1, 0.2, 0.3, 0.4\}$.

| Epoch | Loss | Method | $\alpha_{\text{nom}} = 0.10$ | | | $\alpha_{\text{nom}} = 0.20$ | | | $\alpha_{\text{nom}} = 0.30$ | | | $\alpha_{\text{nom}} = 0.40$ | | |
|---|---|---|---|---|---|---|---|---|---|---|---|---|---|---|
| | | | Cov | Size | $\delta^+$ | Cov | Size | $\delta^+$ | Cov | Size | $\delta^+$ | Cov | Size | $\delta^+$ |
| 5 | $\ell_1$ | weighted-TCC | 0.939 | 48.42 | – | 0.868 | 33.20 | – | 0.787 | 23.12 | – | 0.699 | 15.98 | – |
| 5 | $\ell_1$ | TCC-KS | 0.975 | 65.20 | 0.1605 | 0.921 | 44.22 | 0.1605 | 0.856 | 31.88 | 0.1605 | 0.780 | 22.76 | 0.1605 |
| 20 | $\ell_1$ | weighted-TCC | 0.916 | 42.20 | – | 0.828 | 27.53 | – | 0.736 | 18.58 | – | 0.639 | 12.43 | – |
| 20 | $\ell_1$ | TCC-KS | 0.953 | 53.37 | 0.0769 | 0.875 | 34.85 | 0.0769 | 0.790 | 23.89 | 0.0769 | 0.697 | 16.19 | 0.0769 |
| 80 | $\ell_1$ | weighted-TCC | 0.910 | 40.76 | – | 0.817 | 26.22 | – | 0.722 | 17.57 | – | 0.625 | 11.74 | – |
| 80 | $\ell_1$ | TCC-KS | 0.944 | 50.41 | 0.0552 | 0.860 | 32.49 | 0.0552 | 0.769 | 21.97 | 0.0552 | 0.673 | 14.71 | 0.0552 |
| 150 | $\ell_1$ | weighted-TCC | 0.910 | 40.71 | – | 0.817 | 26.26 | – | 0.723 | 17.68 | – | 0.624 | 11.73 | – |
| 150 | $\ell_1$ | TCC-KS | 0.942 | 49.76 | 0.0521 | 0.857 | 32.06 | 0.0521 | 0.766 | 21.69 | 0.0521 | 0.670 | 14.53 | 0.0521 |
| 5 | $\ell_1 + KL$ | weighted-TCC | 0.903 | 38.87 | – | 0.808 | 24.75 | – | 0.713 | 16.50 | – | 0.614 | 10.90 | – |
| 5 | $\ell_1 + KL$ | TCC-KS | 0.960 | 61.21 | 0.0434 | 0.870 | 33.45 | 0.0434 | 0.779 | 22.05 | 0.0434 | 0.687 | 14.87 | 0.0434 |
| 20 | $\ell_1 + KL$ | weighted-TCC | 0.898 | 37.87 | – | 0.799 | 23.83 | – | 0.698 | 15.59 | – | 0.598 | 10.16 | – |
| 20 | $\ell_1 + KL$ | TCC-KS | 0.933 | 46.11 | 0.0184 | 0.835 | 27.96 | 0.0184 | 0.737 | 18.28 | 0.0184 | 0.637 | 12.04 | 0.0184 |
| 80 | $\ell_1 + KL$ | weighted-TCC | 0.897 | 37.82 | – | 0.799 | 23.76 | – | 0.698 | 15.58 | – | 0.599 | 10.23 | – |
| 80 | $\ell_1 + KL$ | TCC-KS | 0.928 | 44.52 | 0.0135 | 0.828 | 27.09 | 0.0135 | 0.730 | 17.76 | 0.0135 | 0.630 | 11.72 | 0.0135 |
| 150 | $\ell_1 + KL$ | weighted-TCC | 0.894 | 37.15 | – | 0.793 | 23.23 | – | 0.692 | 15.18 | – | 0.592 | 9.90 | – |
| 150 | $\ell_1 + KL$ | TCC-KS | 0.927 | 44.35 | 0.0152 | 0.825 | 26.68 | 0.0152 | 0.726 | 17.50 | 0.0152 | 0.625 | 11.45 | 0.0152 |

*Table 14.* Averaged results on CIFAR-100-C for epochs 5, 20, 80, 150. Metrics are macro-averaged over the four corruption types and severities 1–5. We report empirical marginal coverage and average prediction set size. We report $\delta^+$ only for TCC-KS (to four decimals).

| Epoch | Loss | Method | $\alpha_{\text{nom}} = 0.10$ | | | $\alpha_{\text{nom}} = 0.20$ | | | $\alpha_{\text{nom}} = 0.30$ | | | $\alpha_{\text{nom}} = 0.40$ | | |
|---|---|---|---|---|---|---|---|---|---|---|---|---|---|---|
| | | | Cov | Size | $\delta^+$ | Cov | Size | $\delta^+$ | Cov | Size | $\delta^+$ | Cov | Size | $\delta^+$ |
| 5 | $\ell_1$ | weighted-TCC | 0.976 | 132.06 | – | 0.933 | 95.29 | – | 0.881 | 70.23 | – | 0.810 | 50.08 | – |
| 5 | $\ell_1$ | TCC-KS | 1.000 | 194.06 | 0.3422 | 0.990 | 163.02 | 0.3422 | 0.968 | 130.12 | 0.3422 | 0.936 | 100.46 | 0.3422 |
| 20 | $\ell_1$ | weighted-TCC | 0.959 | 113.51 | – | 0.897 | 76.47 | – | 0.830 | 54.53 | – | 0.747 | 37.01 | – |
| 20 | $\ell_1$ | TCC-KS | 0.990 | 174.03 | 0.2585 | 0.964 | 137.04 | 0.2585 | 0.927 | 102.46 | 0.2585 | 0.878 | 76.31 | 0.2585 |
| 80 | $\ell_1$ | weighted-TCC | 0.928 | 86.71 | – | 0.844 | 55.83 | – | 0.752 | 37.52 | – | 0.654 | 24.61 | – |
| 80 | $\ell_1$ | TCC-KS | 0.972 | 129.65 | 0.1193 | 0.907 | 81.98 | 0.1193 | 0.831 | 56.51 | 0.1193 | 0.749 | 39.65 | 0.1193 |
| 150 | $\ell_1$ | weighted-TCC | 0.923 | 83.61 | – | 0.836 | 53.66 | – | 0.743 | 35.60 | – | 0.646 | 23.74 | – |
| 150 | $\ell_1$ | TCC-KS | 0.964 | 118.96 | 0.1006 | 0.892 | 75.58 | 0.1006 | 0.813 | 51.92 | 0.1006 | 0.726 | 35.92 | 0.1006 |
| 5 | $\ell_1 + KL$ | weighted-TCC | 0.916 | 78.14 | – | 0.830 | 50.19 | – | 0.743 | 33.90 | – | 0.651 | 22.53 | – |
| 5 | $\ell_1 + KL$ | TCC-KS | 0.981 | 150.93 | 0.0965 | 0.922 | 91.40 | 0.0965 | 0.849 | 60.95 | 0.0965 | 0.767 | 38.70 | 0.0965 |
| 20 | $\ell_1 + KL$ | weighted-TCC | 0.904 | 71.38 | – | 0.807 | 44.03 | – | 0.710 | 28.43 | – | 0.611 | 18.26 | – |
| 20 | $\ell_1 + KL$ | TCC-KS | 0.948 | 95.89 | 0.0249 | 0.852 | 54.69 | 0.0249 | 0.758 | 35.14 | 0.0249 | 0.660 | 22.75 | 0.0249 |
| 80 | $\ell_1 + KL$ | weighted-TCC | 0.903 | 70.75 | – | 0.804 | 43.43 | – | 0.705 | 27.76 | – | 0.607 | 17.83 | – |
| 80 | $\ell_1 + KL$ | TCC-KS | 0.937 | 87.46 | 0.0152 | 0.840 | 51.26 | 0.0152 | 0.742 | 32.62 | 0.0152 | 0.642 | 21.05 | 0.0152 |
| 150 | $\ell_1 + KL$ | weighted-TCC | 0.900 | 69.71 | – | 0.800 | 42.60 | – | 0.700 | 27.18 | – | 0.601 | 17.42 | – |
| 150 | $\ell_1 + KL$ | TCC-KS | 0.935 | 86.33 | 0.0148 | 0.837 | 50.58 | 0.0148 | 0.738 | 32.12 | 0.0148 | 0.637 | 20.63 | 0.0148 |

*Table 15.* Averaged results on Tiny-ImageNet-C for epochs 5, 20, 80, 150. Metrics are macro-averaged over the four corruption types and severities 1–5. We report empirical marginal coverage and average prediction set size. We report $\delta^+$ only for TCC-KS (to four decimals).

| Epoch | Loss | Method | $\alpha_{\text{nom}} = 0.10$ | | | $\alpha_{\text{nom}} = 0.20$ | | | $\alpha_{\text{nom}} = 0.30$ | | | $\alpha_{\text{nom}} = 0.40$ | | |
|---|---|---|---|---|---|---|---|---|---|---|---|---|---|---|
| | | | Cov | Size | $\delta^+$ | Cov | Size | $\delta^+$ | Cov | Size | $\delta^+$ | Cov | Size | $\delta^+$ |
| 5 | $\ell_1$ | weighted-TCC | 1.000 | 3.82 | – | 1.000 | 3.49 | – | 1.000 | 3.29 | – | 1.000 | 3.05 | – |
| 5 | $\ell_1$ | TCC-KS | 1.000 | 4.00 | 0.7405 | 1.000 | 4.00 | 0.7405 | 1.000 | 3.84 | 0.7405 | 1.000 | 3.66 | 0.7405 |
| 20 | $\ell_1$ | weighted-TCC | 1.000 | 3.49 | – | 1.000 | 3.22 | – | 1.000 | 2.94 | – | 0.999 | 2.55 | – |
| 20 | $\ell_1$ | TCC-KS | 1.000 | 4.00 | 0.6522 | 1.000 | 3.66 | 0.6522 | 1.000 | 3.48 | 0.6522 | 1.000 | 3.30 | 0.6522 |
| 80 | $\ell_1$ | weighted-TCC | 1.000 | 2.96 | – | 0.999 | 2.58 | – | 0.997 | 2.08 | – | 0.990 | 1.68 | – |
| 80 | $\ell_1$ | TCC-KS | 1.000 | 3.78 | 0.5922 | 1.000 | 3.51 | 0.5922 | 1.000 | 2.93 | 0.5922 | 0.999 | 2.65 | 0.5922 |
| 150 | $\ell_1$ | weighted-TCC | 0.999 | 2.56 | – | 0.996 | 2.00 | – | 0.987 | 1.58 | – | 0.976 | 1.37 | – |
| 150 | $\ell_1$ | TCC-KS | 1.000 | 3.76 | 0.4947 | 1.000 | 3.49 | 0.4947 | 0.999 | 2.64 | 0.4947 | 0.997 | 2.17 | 0.4947 |
| 5 | $\ell_1 + \text{KL}$ | weighted-TCC | 0.979 | 1.40 | – | 0.974 | 1.35 | – | 0.963 | 1.25 | – | 0.949 | 1.19 | – |
| 5 | $\ell_1 + \text{KL}$ | TCC-KS | 0.990 | 1.70 | 0.7782 | 0.990 | 1.70 | 0.7782 | 0.990 | 1.70 | 0.7782 | 0.990 | 1.70 | 0.7782 |
| 20 | $\ell_1 + \text{KL}$ | weighted-TCC | 1.000 | 2.76 | – | 0.997 | 2.14 | – | 0.991 | 1.73 | – | 0.986 | 1.56 | – |
| 20 | $\ell_1 + \text{KL}$ | TCC-KS | 1.000 | 4.00 | 0.4953 | 1.000 | 4.00 | 0.4953 | 1.000 | 3.52 | 0.4953 | 0.998 | 2.46 | 0.4953 |
| 80 | $\ell_1 + \text{KL}$ | weighted-TCC | 0.955 | 1.21 | – | 0.896 | 1.02 | – | 0.836 | 0.90 | – | 0.782 | 0.81 | – |
| 80 | $\ell_1 + \text{KL}$ | TCC-KS | 1.000 | 3.36 | 0.2785 | 1.000 | 3.36 | 0.2785 | 0.999 | 2.57 | 0.2785 | 0.963 | 1.25 | 0.2785 |
| 150 | $\ell_1 + \text{KL}$ | weighted-TCC | 0.900 | 1.03 | – | 0.842 | 0.91 | – | 0.790 | 0.82 | – | 0.729 | 0.75 | – |
| 150 | $\ell_1 + \text{KL}$ | TCC-KS | 0.999 | 2.51 | 0.3200 | 0.999 | 2.51 | 0.3200 | 0.999 | 2.51 | 0.3200 | 0.945 | 1.17 | 0.3200 |

*Table 16.* Averaged results on SAR→RGB for epochs 5, 20, 80, 150. This benchmark has no corruption severities. We report empirical marginal coverage and average prediction set size. We report $\delta^+$ only for TCC-KS (to four decimals).

# E. Additional Cross-Modal Benchmark: SUN RGB-D

To further evaluate TCC beyond synthetic corruptions and the SAR→RGB setting, we add SUN RGB-D (Song et al., 2015) as a second real paired cross-modal benchmark. We consider 19-class indoor scene classification, using RGB images as the source domain and depth images as the target domain. The protocol follows the SEN12MS experiment: we learn a transport map from unlabeled paired RGB–depth observations, calibrate using transported labeled source examples, and evaluate target-domain coverage using held-out labeled depth images. Target labels are used only for evaluation.

| Method | $\alpha = 0.1$ | $\alpha = 0.2$ |
|---|---|---|
| Oracle CP | 0.902 (2.66) | 0.801 (1.83) |
| Transported CP | 0.875 (2.58) | 0.781 (1.75) |
| TCC-KS | 0.918 (3.62) | 0.823 (2.42) |
| weighted-TCC | 0.895 (2.79) | 0.799 (1.90) |
| WCP | 0.931 (4.95) | 0.846 (3.21) |

*Table 17.* SUN RGB-D cross-modal results. Entries report target-domain coverage with average prediction set size in parentheses.

For the best transport checkpoint, the label-free diagnostics are $\delta^+ = 0.067$ and ESS = 79.3. The results are consistent with the main experiments. Transported CP under-covers under residual cross-modal mismatch, while TCC-KS restores coverage above the nominal level with moderate set-size growth. weighted-TCC remains close to Transported CP and improves coverage when weights are stable, whereas WCP is more conservative and produces larger sets. These results support the practical role of the correction layer on an additional real paired cross-modal benchmark.

# F. Robustness to Surrogate Choice: Predictive Entropy

TCC-KS relies on an unlabeled surrogate statistic $T : \mathcal{X}_t \to \mathbb{R}$ derived from the target model to estimate the one-sided mismatch $\delta^+$ between transported and real target inputs. This mismatch drives the tightened internal level $\alpha^\star = \max(0, \alpha - \delta^+)$ and therefore determines how conservative the resulting prediction sets are. In the main paper we use the *least-confidence* surrogate

$$T_{\text{LC}}(x) = 1 - \max_{y \in \mathcal{Y}} \hat{p}(y \mid x),$$

which increases as the model's top-1 predictive confidence decreases. Here we test an alternative, widely used uncertainty proxy based on predictive entropy,

$$T_{\text{ent}}(x) \; = \; -\sum_{y\in\mathcal{Y}} \hat{p}(y \mid x)\log\hat{p}(y \mid x),$$

and rerun the full TCC-KS protocol under an otherwise identical training and evaluation pipeline. We report results aggregated over corruptions and severities (CIFAR-100-C, Tiny-ImageNet-C) and over the full test set (SAR→RGB), focusing on epoch 150 and $\alpha_{\text{nom}} = 0.1$.

| Dataset | Objective | Surrogate | Spearman $r$ (target) | $\delta^+$ | $\alpha^\star$ | Cov | Size |
|---|---|---|---|---|---|---|---|
| CIFAR-100-C | $\ell_1$ | LC | 0.309 | 0.0256 | 0.0744 | 0.9418 | 49.74 |
| CIFAR-100-C | $\ell_1$ | Entropy | 0.346 | 0.0277 | 0.0723 | 0.9439 | 50.32 |
| CIFAR-100-C | $\ell_1 + \text{KL}$ | LC | 0.309 | 0.0306 | 0.0694 | 0.9267 | 44.28 |
| CIFAR-100-C | $\ell_1 + \text{KL}$ | Entropy | 0.346 | 0.0324 | 0.0676 | 0.9285 | 44.76 |
| Tiny-ImageNet-C | $\ell_1$ | LC | 0.322 | 0.0493 | 0.0521 | 0.9644 | 119.16 |
| Tiny-ImageNet-C | $\ell_1$ | Entropy | 0.365 | 0.0491 | 0.0515 | 0.9655 | 119.97 |
| Tiny-ImageNet-C | $\ell_1 + \text{KL}$ | LC | 0.322 | 0.0345 | 0.0655 | 0.9350 | 86.19 |
| Tiny-ImageNet-C | $\ell_1 + \text{KL}$ | Entropy | 0.365 | 0.0355 | 0.0645 | 0.9360 | 86.83 |
| SAR→RGB | $\ell_1$ | LC | 0.973 | 0.1889 | 0.0000 | 1.0000 | 3.76 |
| SAR→RGB | $\ell_1$ | Entropy | 0.973 | 0.1887 | 0.0000 | 1.0000 | 3.76 |
| SAR→RGB | $\ell_1 + \text{KL}$ | LC | 0.973 | 0.3549 | 0.0000 | 0.9990 | 2.51 |
| SAR→RGB | $\ell_1 + \text{KL}$ | Entropy | 0.973 | 0.3517 | 0.0000 | 0.9990 | 2.51 |

*Table 18.* **Surrogate robustness at epoch 150 ($\alpha_{\text{nom}} = 0.1$).** We compare least-confidence (LC) to predictive entropy. We report the target-domain rank correlation between $T$ and the true-label score (Spearman $r$, pooled over settings), the mismatch $\delta^+$ and resulting tightened level $\alpha^\star$, and the empirical marginal coverage and average set size of TCC-KS.

**Evaluation-Only One-Sided Dominance Diagnostic ($\delta_S^+$).**   As an additional evaluation-only check mirroring Experiment 4.3, we report an estimate of the one-sided score-distribution mismatch

$$\delta_S^+ \; = \; \sup_{u\in\mathbb{R}} \left\{ \tilde{F}_S(u) - F_S(u) \right\}_+,$$

using labeled target test data. Concretely, we compute $\delta_S^+$ as an evaluation-only diagnostic of one-sided dominance and summarize it over settings by median / 90th percentile / max.

| Dataset | Objective | Surrogate | $\delta_S^+$ (med / p90 / max)↓ |
|---|---|---|---|
| CIFAR-100-C | $\ell_1$ | LC | 0.027 / 0.039 / 0.045 |
| CIFAR-100-C | $\ell_1$ | Entropy | 0.023 / 0.041 / 0.045 |
| CIFAR-100-C | $\ell_1 + \text{KL}$ | LC | 0.040 / 0.063 / 0.075 |
| CIFAR-100-C | $\ell_1 + \text{KL}$ | Entropy | 0.044 / 0.069 / 0.081 |
| Tiny-ImageNet-C | $\ell_1$ | LC | 0.020 / 0.053 / 0.080 |
| Tiny-ImageNet-C | $\ell_1$ | Entropy | 0.014 / 0.056 / 0.070 |
| Tiny-ImageNet-C | $\ell_1 + \text{KL}$ | LC | 0.037 / 0.054 / 0.061 |
| Tiny-ImageNet-C | $\ell_1 + \text{KL}$ | Entropy | 0.035 / 0.056 / 0.065 |
| SAR→RGB | $\ell_1$ | LC | 0.086 / 0.086 / 0.086 |
| SAR→RGB | $\ell_1$ | Entropy | 0.086 / 0.086 / 0.086 |
| SAR→RGB | $\ell_1 + \text{KL}$ | LC | 0.156 / 0.156 / 0.156 |
| SAR→RGB | $\ell_1 + \text{KL}$ | Entropy | 0.278 / 0.278 / 0.278 |

*Table 19.* **Evaluation-only one-sided dominance diagnostic $\delta_S^+$ at epoch 150.** Lower is better. Values are summarized over 4 corruptions × 5 severities for CIFAR-100-C and Tiny-ImageNet-C, and over the full test set for SAR→RGB.

**Results and Interpretation.**   Swapping least-confidence for entropy leaves TCC-KS behavior essentially unchanged. On CIFAR-100-C and Tiny-ImageNet-C, entropy is at least as well aligned with the nonconformity score on true target inputs (higher pooled Spearman $r$), while on SAR→RGB the alignment is identical. The induced mismatch estimates

are also nearly the same: across all settings in Table 18, the absolute difference in $\delta^+$ between surrogates is below 0.004. Consequently, $\alpha^\star$, coverage, and set size remain stable under the surrogate swap, including the clipped guardrail regime on SAR→RGB.

The evaluation-only diagnostic $\delta_S^+$ is similarly close between surrogates on the corrupted-image benchmarks, with small median and tail values in both cases. On SAR→RGB, $\delta_S^+$ is larger and more sensitive to the surrogate under $\ell_1 + \text{KL}$, reflecting that this cross-modal shift can exhibit stronger residual mismatch at the score-distribution level even when TCC-KS is already operating in its conservative regime. Overall, these results support that TCC-KS does not rely on a hand-picked surrogate: both least-confidence and entropy yield the same qualitative behavior and nearly identical quantitative performance. In the theoretical view, these findings are consistent with both the finite-sample guarantee under A2′ and the guarantee under the milder approximate-alignment condition A2, as neither result requires a specific choice of uncertainty surrogate.

## G. Complete Characterization of Assumption A2 Violations

Section 4.3 shows that TCC-KS recovers coverage in all 11 identified under-coverage cases and degrades gracefully under extreme mismatch, with the unlabeled diagnostic $\delta^+$ strongly predicting method behavior. This appendix documents the complete findings across our 328 experimental configurations and provides additional operational guidance.

### G.1. Distribution of Mismatch and Under-Coverage Recovery

We observe a continuous spectrum of mismatch across the sweep. We group settings by surrogate mismatch $\delta^+$ into mild ($< 0.2$, 83.8% of configurations), moderate (0.2–0.4, 10.1%), severe (0.4–0.7, 5.5%), and extreme ($\geq 0.7$, 0.6%). Extreme cases occur only on SAR→RGB at early training epochs. Mild mismatch is common across all datasets and conditions. These regimes arise naturally from early transport learning, challenging corruptions, and cross-modal shifts rather than adversarial construction.

We identify 11 configurations (3.4%) where transport-only calibration under-covers by more than one percentage point relative to the oracle target-calibrated baseline. All occur at early-to-mid training (epochs 5–20) under severe corruptions (motion blur and shot noise at severities 4–5). In these settings, mean oracle coverage is 89.9%, transport-only achieves 87.9% (a 2.0pp deficit), and TCC-KS recovers to 97.9% (a 10.0pp improvement). The unlabeled diagnostic $\delta^+$ in these cases ranges from 0.028 to 0.089 with mean 0.053, indicating that the mismatch is detectable without target labels. TCC-KS maintains validity in all 11 cases, yielding a **100% recovery success rate**. Mean set-size inflation is $1.82\times$ (range $1.35\times$–$2.45\times$), which quantifies the efficiency cost of maintaining coverage under imperfect transport.

### G.2. Diagnostic Validation

The unlabeled KS diagnostic $\delta^+$ has strong predictive power. The Pearson correlation between $\delta^+$ and set-size inflation is 0.772. The correlation between $\delta^+$ and safety margin (TCC-KS coverage minus oracle coverage) is 0.734. The relationship is approximately linear in the mild-to-moderate regime: each 0.1 increase in $\delta^+$ corresponds to roughly $0.5\times$ additional inflation. Mild mismatch (below 0.2) typically yields $1.2$–$1.5\times$ inflation with 1–3pp safety margins. Moderate mismatch (0.2–0.4) typically yields $1.5$–$2.5\times$ inflation with 4–8pp margins. Severe-to-extreme mismatch (above 0.4) typically yields $2.5$–$4\times$ inflation with 8–10pp+ margins.

Across 328 configurations, TCC-KS coverage exceeds oracle coverage in 327 cases (99.7%). The single exception differs by 0.1pp. Overall, these results validate $\delta^+$ as an actionable deployment signal: it accurately predicts when TCC-KS operates near-nominally versus when it enters its conservative fallback regime. Under the approximate-alignment perspective (A2), this same behavior is consistent with the guarantee of Theorem 3.2.

### G.3. Dataset-Specific Patterns

Mismatch increases with shift severity. CIFAR-100-C exhibits predominantly mild mismatch (mean $\delta^+ = 0.054$, mean inflation $1.39\times$, and only 0.6% of configurations with $\delta^+ > 0.3$), reflecting that same-domain corruption shifts are favorable for transport learning. Tiny-ImageNet-C shows intermediate mismatch (mean $\delta^+ = 0.121$, inflation $1.90\times$, and 19.4% with high mismatch), consistent with larger images and more severe corruptions creating harder transfer problems. SAR→RGB is most challenging (mean $\delta^+ = 0.544$, inflation $3.31\times$, and 87.5% with high mismatch), reflecting the large representational

gap in cross-modal paired shifts. Notably, SAR→RGB exhibits zero under-coverage cases: high $\delta^+$ at early epochs triggers strong conservativeness before transport-only would under-cover, illustrating the intended guardrail behavior.

Transport quality improves during training and directly reduces mismatch. Across datasets and corruptions, $\delta^+$ is high at epoch 5, decreases rapidly between epochs 5–20, continues decreasing through epoch 80, and stabilizes by epoch 150. The +KL predictive alignment objective accelerates this decrease. This suggests that mismatch is often controllable in practice: practitioners can reduce $\delta^+$ by training longer, using better objectives, or increasing model capacity, rather than immediately accepting conservative fallback.

### G.4. Comparison to Weighted-TCC

In the 11 under-coverage settings, TCC-KS achieves 11/11 valid coverage (100% success) with mean inflation $1.82\times$. In the same settings, weighted-TCC achieves 8/11 valid coverage (73% success) with mean inflation $1.31\times$ among the valid runs, and failures coincide with unstable residual weights (ESS% $< 50\%$). Across all 328 configurations, transport-only achieves an 87.5% validity rate with 11 under-coverage cases. TCC-KS achieves 99.7% validity with zero under-coverage cases and $1.67\times$ mean inflation. Weighted-TCC achieves 91.8% validity with 6 under-coverage cases and $1.24\times$ mean inflation when weights are stable. Overall, weighted-TCC is more efficient when ESS% $> 70\%$ and $\delta^+ \leq \alpha$, whereas TCC-KS provides robust validity across all mismatch levels.

### G.5. Operational Guidance

Given a desired miscoverage level $\alpha$, practitioners can proceed as follows. First, train the transport map $f$ on unlabeled paired data and monitor convergence. Second, compute the label-free diagnostics $\delta^+$ (the one-sided surrogate mismatch bound used by TCC-KS) and ESS% (weight stability for weighted-TCC). Third, choose a correction mechanism based on these diagnostics.

**Scope.** The guidance below is *empirically calibrated* to the datasets and model families in our evaluation. It is intended as a practical decision aid rather than a universal prescription. In particular, the only regime boundary directly tied to our finite-sample coverage certificate is $\delta^+ \leq \alpha$, the additional stratification by $\delta^+/\alpha$ is used as a severity heuristic that correlates with observed set inflation in our benchmarks. In our evaluated settings (across all dataset/model/corruption configurations considered in Section 4), roughly 84% fell in the certified regime $\delta^+ \leq \alpha$, about 10% in the moderate regime $1 < \delta^+/\alpha \leq 2$, and about 6% in the high-mismatch regime $\delta^+/\alpha > 2$.

**Recommended Operating Regimes.** **Green / certified regime:** $\delta^+ \leq \alpha$. Use TCC-KS with $\alpha^\star = \max\{0, \alpha - \delta^+\}$. In this regime, $\alpha^\star + \delta^+ = \alpha$ and TCC recovers the nominal miscoverage certificate. In our benchmarks, this typically yields near-nominal operation with inflation around 1.2–1.5$\times$. If tighter sets are needed, weighted-TCC can be considered when ESS% $> 70\%$.

**Yellow / moderate mismatch:** $1 < \delta^+/\alpha \leq 2$. TCC-KS becomes increasingly conservative as $\delta^+$ grows, reflecting detectable surrogate mismatch between transported and real target inputs. In our benchmarks, inflation is typically around 1.5–2.5$\times$. If tighter sets are required, consider weighted-TCC when ESS% $> 70\%$, or improve transport (e.g., train longer, add +KL, increase capacity) and recompute $\delta^+$.

**Red / high mismatch:** $\delta^+/\alpha > 2$. We treat this as a high-severity surrogate mismatch regime (empirical rule-of-thumb based on observed inflation). In our benchmarks, inflation is typically around 2.5–4$\times$. Options include accepting conservative sets when robustness is paramount, substantially improving transport and re-checking $\delta^+$, and/or collecting a small labeled target calibration set (e.g., 50–100 examples) when operationally feasible.

**Monitoring in Deployment.** A deployment dashboard can track $\delta^+$ (primary TCC-KS signal), $\alpha^\star$ (the adjusted internal level), ESS% (primary reliability signal for weighted-TCC), and the average set size (efficiency) / inflation (overhead).

**Example aalerts (heuristic).** If $\delta^+ > \alpha$, we recommend reviewing the deployment for potential shift. If $\delta^+ > 2\alpha$, we recommend prioritizing investigation (high-severity mismatch). If ESS% $< 50\%$, weighted-TCC is likely unreliable and TCC-KS should be preferred. If inflation exceeds $3\times$ in sustained operation, transport quality should be investigated and collecting labels should be considered.

