# OpenReview forum: "Conformal Calibration Transfer"
_ICML.cc/2026/Conference — ICML 2026 regular_

### Official Review · Reviewer_yu77 · 2026-02-19

**Soundness:** 2
**Presentation:** 1
**Significance:** 1
**Originality:** 2
**Overall Recommendation:** 4
**Confidence:** 1

**Summary:**

This research focuses on conformal calibration transfer where the source domain has labeled calibration sets and the target domain has only unlabeled data but paired observations (such as modal/sensor variations) with the source domain. It proposes Transported Conformal Calibration (TCC): first, paired data is used to transport labeled calibration samples from the source domain to the target domain; then, unlabeled input from only the target domain is used to correct the residual distribution mismatch after transport. Two implementations are presented: TCC-KS (using label-free uncertainty surrogate for mismatch detection and conservative adjustment) and weighted-TCC (reweighting for improved efficiency). The research also provide a finite-sample target domain coverage guarantee, which is validated on CIFAR-100-C, Tiny-ImageNet-C, and SEN12MS.

**Compliance With Llm Reviewing Policy:**

Affirmed.

**Final Justification:**

Satisfy on rebuttal, Increse from 3 to 4.

**Key Questions For Authors:**

1. See Weakness.
2. If you can characterize the mismatch with observables (only the target domain is unlabeled) and incorporate it into threshold/quantile correction, this is very similar to the correct approach to conformalism: trading extra conservatism for target-domain validity.
3. Can you plot an architect(or Workflow description) to help us unstanding your methods?
4. How exactly is the "transport labeled calibration" step performed?
5. Is the surrogate sufficiently discriminative? Is it insensitive to certain shifts?
6. Are "target-domain coverage guarantees" overall marginal coverage, or some kind of conditional/grouped coverage? Is per-corruption type coverage reported on the corruption benchmark?

**Limitations:**

1. Guarantees is marginal, not conditional validity. Standard conformal prediction guarantees overall (marginal) coverage, but it can still under-cover for particular inputs or subgroups; distribution-free conditional coverage is generally impossible without overly large sets.
2. Maybe fragile under distribution shift. The exchangeability/i.i.d. assumption is crucial—under covariate shift, label shift, corruption, or OOD deployment, coverage guarantees can break and empirical coverage may drop.
3. Efficiency can be poor. Even if coverage holds, prediction sets/intervals may become very large (wide intervals or many labels), especially with weak base models, high noise, large-class problems, or stronger shifts—reducing decision usefulness.

**Strengths And Weaknesses:**

Strengths:
1. The problem is clearly defined and realistic: the target domain has no labels, but paired data (corresponding samples between domains) exists, which is very common in cross-sensor/cross-modal/style transfer scenarios. Clearly defining "conformalism that cannot satisfy exchangeability" as a calibration transfer problem is reasonable.
2. The method structure is reasonable: transport + mismatch correction. Performing transport first and then correction essentially addresses whether the residual distribution after transport remains consistent with the target domain.
3. Claiming to offer a finite-sample target coverage guarantee is a key selling point. If proven true and the assumptions are clear, this is far stronger than simply stating "experienced coverage is good."

Weaknesses:
1. The phrase "target space linked to source through unlabeled paired observations" in the abstract is crucial but vague. Different mechanisms for generating paired observations lead to vastly different theories: Is it the same object observed simultaneously by two sensors (strong correspondence)? Or is it only weak pairing/pseudo-pairing/temporally proximate pairing? Are paired observations i.i.d.? Is there selection bias? These directly determine the "transport effectiveness" and the reliability of the coverage guarantee.
2. Different transports introduce different sources of error: Approximation error, Residual distribution drift (post-transport mismatch). And the sensitivity of the calibration score to the mapping (e.g., logit-based vs. feature-based).It doesn't clearly state whether the "transport object" is the input, representation, or nonconformity score.
3. The TCC-KS "label-free uncertainty surrogate" might be unstable. It will be better if we could see the criteria used to select the surrogate and its robustness to different shift types.
4. What is the measure of residual mismatch after transport? How does mismatch occur in theoretical guarantees? Is it additive slack or data-dependent bound?

---

> ### Author Rebuttal · Authors · 2026-03-30
>
> We thank the reviewer for the careful reading and for identifying several places where the paper can be made clearer. Below, we address the main concerns.
> ## Weakness 1 — paired observations
> The intended setting is the instance-level paired regime: each pair $(x_s,x_t)$ is the same underlying instance observed in two input spaces, with paired data modeled as i.i.d. draws $(x_s^{(j)},x_t^{(j)})\sim P_{X_sX_t}$. Typical examples are co-registered measurements or synchronized captures. So the theory does **not** cover arbitrary pseudo-pairing, loose temporal proximity, or selection-biased pairing.
> ## Weakness 2 — what is transported
> The transported object is the **input data**: $f:X_s\to X_t$. We form $\widetilde x_t^{(i)}=f(x_s^{(i)})$, keep $y^{(i)}$ unchanged, and compute the **target-side** score $S_t(\widetilde x_t^{(i)},y^{(i)})$ with the fixed target predictor $g_t$. So we do **not** transport features or scores directly; we transport inputs into the target space and then use the same target-side score as at deployment.
> ## Weakness 3 — surrogate robustness
> On how the surrogate was chosen and whether it is robust, our selection criterion was intentionally simple: the surrogate should (i) be **computable without target labels** from the deployed target predictor alone, and (ii) track the model’s notion of **difficulty / uncertainty** in a way plausibly coupled to the true-label score. This is why our main choice is least confidence, $T(x)=1-\max_y \hat p_t(y\mid x)$.
>
> Its robustness is already checked in Appendix E by replacing least confidence with predictive entropy and rerunning the full protocol. The behavior is nearly unchanged: e.g., on CIFAR-100-C, $\delta^+$ changes from 0.0306 to 0.0324 and coverage from 0.9267 to 0.9285. Appendix C further shows that the behavior remains stable across corruption types and severities.
>
> So, for the regimes studied, the surrogate is selected by clear label-free criteria and is empirically robust across both alternative surrogate choices and different shift types.
> ## Response to Weakness 4 — residual mismatch in the guarantee
> Residual mismatch appears through **two terms**: the data-dependent certificate $\delta^+$, estimated from unlabeled target and transported inputs, and the additive slack $\varepsilon$, which appears in Assumption A2 via $\delta_S \le \delta_T + \varepsilon$.
> ## Question 2 — observable mismatch
> Yes, if one can characterize mismatch with observables and incorporate it into a quantile correction, this is the right conformal mechanism.
> However, **this setup is very different from ours**, as we note in L056 left column: in our problem, the true score-side mismatch is not observable because the target domain is unlabeled. This is exactly why our impossibility result (Theorem 3.1) matters: without additional structure, nontrivial label-free transfer is impossible in general. Our contribution is to show that, under Assumption A2, an **observable surrogate-side mismatch** can still be turned into a finite-sample correction through $\delta^+$, by setting $\alpha^\star=\max(0,\alpha-\delta^+)$.
> ## Question 3 — workflow
> We will add a workflow diagram in the revised manuscript. For convenience, the workflow is:
> 1. learn $f:X_s\to X_t$ from unlabeled paired data;
> 2. transport labeled source calibration inputs: $\widetilde x_t^{(i)}=f(x_s^{(i)})$, keeping labels unchanged;
> 3. compute target-side scores $S_t(\widetilde x_t^{(i)},y^{(i)})$;
> 4. apply either TCC-KS (estimate $\delta^+$ and tighten calibration) or weighted-TCC (estimate residual density ratios and reweight);
> 5. output target-space prediction sets for the fixed predictor $g_t$.
> ## Question 4 — transport calibration
> The transported-calibration step is simply: $\widetilde x_t^{(i)}=f(x_s^{(i)})$, keep $y^{(i)}$ unchanged, and calibrate with $S_t(\widetilde x_t^{(i)},y^{(i)})$.
> ## Question 5 — is the surrogate sufficiently discriminative?
> For the regimes we study and on the tested benchmarks, yes, the surrogate is sufficiently discriminative for TCC-KS to work as intended, although we do not claim it is universally perfect.
>
> Appendix E studies the surrogate statistic and finds robust behavior across two surrogates: e.g., on CIFAR-100-C, $\delta^+$ changes from 0.0306 to 0.0324 and coverage from 0.9267 to 0.9285. Table 5 complements this with the surrogate–score coupling audit via $\rho(T,S)$.
> ## Question 6 — marginal/conditional coverage
> Our guarantees are marginal coverage guarantees, i.e. the standard split-conformal notion in our theorems. Conditional/group-based coverage is largely orthogonal to our transport-and-correction layer and could, in principle, be combined with it. Per-corruption and per-severity results are already reported in Appendix C, to which we refer in the main paper in L262, L277 right column.
>
> We thank the reviewer again for the thoughtful feedback and hope these clarifications address the concerns. If so, we would be grateful for a reconsideration of the score.

---

> > ### Author Rebuttal · Reviewer_yu77 · 2026-04-01
> >
> > Thank you for the detailed rebuttal. The clarifications are helpful, especially on the paired-observation setting, the transported object, and the residual mismatch terms in the guarantee. In particular, the more precise description of the paired-data assumption also makes the scope of applicability narrower than the paper’s broader framing initially suggested. Authors can also consider about the practical robustness of the approach under varied real-world shifts, especially given its dependence on transport quality and surrogate-based mismatch estimation. Overall, I appreciate the authors’ response, and I will consider adjust my score based on the rebuttal responses on other reviewers. Thanks.

---

> > > ### Author Response · Authors · 2026-04-05
> > >
> > > We thank the reviewer very much for the thoughtful feedback and follow-up. We are glad that our response helped clarify the main points. If the paper is accepted, we will improve the presentation to make these points clearer in the final version.

---

### Official Review · Reviewer_zdBJ · 2026-03-09

**Soundness:** 3
**Presentation:** 3
**Significance:** 3
**Originality:** 4
**Overall Recommendation:** 4
**Confidence:** 3

**Summary:**

This paper studies conformal calibration transfer: labeled calibration data exist only in a source space X_s, but prediction sets are needed in a target space X_t linked to X_s via unlabeled paired observations (e.g., sensor upgrades, modality changes). The authors propose Transported Conformal Calibration (TCC), which first transports labeled source calibration into X_t using a learned map f, then corrects residual post-transport mismatch using only unlabeled target inputs. Two correction mechanisms are proposed: TCC-KS, which forms a label-free mismatch certificate δ+ via a one-sided KS discrepancy on a surrogate statistic and tightens the calibration level accordingly, and weighted-TCC, which reweights calibration scores via density ratios between the transported and true target distributions. An impossibility theorem establishes that label-free coverage transfer without structural assumptions must be near-vacuous, motivating Assumption A2 (approximate surrogate control). Finite-sample target coverage bounds are derived for TCC-KS. Experiments on CIFAR-100-C, Tiny-ImageNet-C, and SAR→RGB (SEN12MS) show reliable coverage transfer without labeled target data, with label-free diagnostics that predict when correction is needed.

**Compliance With Llm Reviewing Policy:**

Affirmed.

**Final Justification:**

The author has solved all my concerns, and I recommend to accept this work.

**Key Questions For Authors:**

1.Assumption A3 requires splitting unlabeled paired data into disjoint subsets for transport learning, KS estimation, and importance weight estimation. How sensitive are δ+ and ESS% to the sizes of these splits when the total budget m is small? Is there empirical guidance (or a rule of thumb) on the minimum m needed for reliable operation of each correction mechanism?

2. A natural baseline in settings where pretrained multimodal encoders exist (e.g., CLIP-style models that embed both modalities into a shared space) would be to apply standard importance-weighted CP directly on nonconformity scores computed in that shared embedding space, bypassing explicit transport learning. Were there practical reasons this was not included as a comparison, and how would TCC be expected to compare in such settings?

**Limitations:**

The paper honestly addresses its main limitations in Section 5: the method requires unlabeled paired observations linking source and target (limiting applicability to settings with synchronized data collection), coverage guarantees are marginal rather than conditional, Assumption A2 cannot be verified without target labels, and explicit sample splitting reduces effective sample size in small-data regimes. These are acknowledged clearly. No additional limitations concerns beyond those already discussed.

**Strengths And Weaknesses:**

**Strengths:**

- The problem formulation is fresh and practically motivated. Cross-space conformal calibration transfer—where source and target inputs live in genuinely different spaces connected only by unlabeled paired observations—has not been addressed in the conformal prediction literature to my knowledge. The running examples (sensor upgrade, modality change) are concrete, and the distinction from standard covariate-shift CP (which assumes a common input space) is clearly drawn.

- The theoretical development is coherent and principled. Theorem 3.1 (impossibility) cleanly motivates why structural assumptions are necessary, and Theorem 3.2 provides an explicit finite-sample coverage bound tied to the deployable observable δ+. Making Assumption A2 explicit and connecting it to a measurable diagnostic is more rigorous than typical in this area.

- The label-free diagnostics (δ+ for TCC-KS, ESS% for weighted-TCC) are practically valuable. The 328-configuration sweep in Appendix F provides strong empirical support: TCC-KS restores valid coverage in all 11 identified under-coverage cases, and δ+ correlates 0.772 with set-size inflation, confirming it as a reliable deployment signal.

**Weaknesses:**

- Assumption A2 (δ_S ≤ δ_T + ε) is the key enabling condition but cannot be verified without target labels. The empirical slack ε_{eval} is small within the tested benchmarks, but the paper provides no guidance on when A2 might fail in new settings or how practitioners could screen for potential violations before deployment.

- When δ+ is large (e.g., SAR→RGB with most transport checkpoints), TCC-KS collapses to α* = 0 and outputs near-vacuous prediction sets. This is technically valid but practically useless in exactly the scenarios that motivate the work. The paper frames this as graceful degradation but does not discuss fallback strategies or criteria for concluding that calibration transfer is infeasible.

- The experimental scope is limited to image classification under synthetic corruptions and a single cross-modal pair. Evaluations on regression tasks, text or audio modalities, or settings where the source and target classifiers differ would better support the claimed generality of the framework.

---

> ### Author Rebuttal · Authors · 2026-03-30
>
> We thank the reviewer for the careful and constructive review, and for highlighting the novelty of the setting, the role of the impossibility result, and the value of the label-free diagnostics. Below, we address the concerns in the same order as the review.
>
> ## Response to Weakness 1 — when A2 may fail / how to screen
> We agree that this is a genuine current limitation. At the same time, in light of our impossibility result (Theorem 3.1), we believe this limitation is fundamentally unavoidable in the label-free setting: without additional structure, one cannot hope for a nontrivial distribution-free transfer guarantee. The relevant practical question is therefore not whether A2 can be directly verified, but whether one can screen for likely failure modes before deployment using only observable quantities. Appendix F already provides this characterization. Across 328 configurations, only 11 (3.4%) show transport-only under-coverage, and all 11 occur in the hardest regimes: early-to-mid transport training and severe corruptions (motion blur / shot noise, severities 4–5). In these cases, oracle coverage is 89.9%, transport-only drops to 87.9%, and TCC-KS recovers to 97.9%. Thus, the empirical picture is that A2-related failure is not diffuse; it appears when residual mismatch after transport remains substantial. The intended screening signal is the observable certificate $\delta^+$, together with ESS for weighted-TCC. Appendix F.1/F.5 already gives the operating rule: use TCC-KS in the certified regime $\delta^+\le\alpha$; if $1<\delta^+/\alpha\le2$, expect increasing conservativeness and prefer weighted-TCC only when ESS is high; if $\delta^+/\alpha>2$, treat this as a high-mismatch warning and either improve transport, fall back to conservative sets, or obtain a small labeled target calibration set. In our audit, about 84% of configurations fall in the certified regime, about 10% in the moderate regime, and about 6% in the high-mismatch regime.
>
> ## Response to Weakness 2 — large-$\delta^+$ / fallback
> We agree that the large-$\delta^+$ regime should be stated more explicitly. Here, TCC-KS is not meant to be an efficient solution to the hardest cases; it is a guardrail. When $\delta^+$ stays large, the right conclusion is that informative label-free transfer is not currently supported by the observable evidence. Appendix F.5 already gives the intended fallback: if $\delta^+$ decreases as transport improves, improve transport; if $\delta^+$ stays large and ESS is also low, then neither TCC-KS nor weighted-TCC should be expected to be practically attractive, and transfer is currently infeasible without stronger structure or some labeled target data.
>
> ## Response to Weakness 3
> To strengthen the empirical scope, we added **SUN RGB-D** as a second real paired cross-modal benchmark (19-class indoor scene classification, RGB$\to$depth). using RGB images as source and depth images as target, with the same protocol as in SEN12MS.
>
> | Method | $\alpha=0.1$ | $\alpha=0.2$ |
> |---|---:|---:|
> | Oracle CP | 0.902 (2.66) | 0.801 (1.83) |
> | Transported CP | 0.875 (2.58) | 0.781 (1.75) |
> | TCC-KS | 0.918 (3.62) | 0.823 (2.42) |
> | weighted-TCC | 0.895 (2.79) | 0.799 (1.90) |
> | WCP | 0.931 (4.95) | 0.846 (3.21) |
>
> For the best transport checkpoint, $\delta^+=0.067$ and $\mathrm{ESS}=79.3\%$. The results remain consistent with the rest of the paper: TCC-KS slightly exceeds nominal coverage with modest set-size growth, while WCP is more conservative; weighted-TCC stays close to Transported CP and efficiently closes most of the gap.
>
> ## Response to Key Question 1
> At a high level, the two corrections behave differently when $m$ is small. For TCC-KS, the finite-sample inflation in $\delta^+$ inherits the DKW terms, so smaller splits make the method noisier and more conservative. For weighted-TCC, the key issue is weight concentration, which is why ESS is the relevant diagnostic. The practical rule-of-thumb, already consistent with Appendix F.4/F.5, is: **TCC-KS is the safer default when $m$ is limited**, whereas **weighted-TCC should be preferred only when ESS indicates a stable weighting regime**.
>
> ## Response to Key Question 2
> We agree this is an interesting baseline family, but it addresses a somewhat different regime. Our setting assumes a fixed target predictor $g_t$ in the target input space $X_t$, and TCC is designed to transfer calibration into that same space so that the conformal score remains tied to the deployed model. A shared-embedding baseline instead assumes the existence of a strong multimodal representation that already aligns the modalities in a score-relevant way. We therefore view this as **complementary rather than contradictory** to TCC, and we will clarify this in the revision and present it as an important future comparison.
>
> We are once again thankful for the constructive feedback, and hope that our answers addressed the reviewer's concerns.

---

> > ### Author Rebuttal · Reviewer_zdBJ · 2026-04-03
> >
> > The author has solved my concern, I will keep my score.

---

> > > ### Author Response · Authors · 2026-04-05
> > >
> > > Thank you very much for your positive evaluation and for your follow-up. We are glad that our rebuttal addressed the concern. We also appreciate your careful reading and constructive feedback. If the paper is accepted, we will incorporate the necessary modifications based on your feedback into the final version.

---

### Official Review · Reviewer_6gcV · 2026-03-10

**Soundness:** 2
**Presentation:** 3
**Significance:** 2
**Originality:** 2
**Overall Recommendation:** 4
**Confidence:** 4

**Summary:**

This paper introduces Transported Conformal Calibration, an approach for transferring conformal prediction guarantees from a labeled source domain to an unlabeled target domain. The core idea is to learn a transport map $f$ from unlabeled paired observations linking both domains, translating labeled source calibration samples into the target space. Then, the residual post-transport mismatch is corrected using two complementary techniques. TCC-KS uses a one-sided Kolmogorov-Smirnov test to produce a mismatch certificate $\delta^+$ and then adjust calibration conservatively. In contrast, weighted-TCC corrects residual mismatch by reweighting these scores toward the real target input distribution. The authors evaluate the approach on CIFAR-100-C, Tiny-ImageNet-C, and SEN12MS, demonstrating reliable coverage transfer without labeled target calibration data.

**Compliance With Llm Reviewing Policy:**

Affirmed.

**Final Justification:**

After reviewing the authors' rebuttal and the discussion, I increase my score from 3 to 4.

**Key Questions For Authors:**

Q1: The method assumes access to perfectly paired observations where each pair corresponds to the same underlying instance. How sensitive is TCC to imperfect or approximate pairing? Could the authors provide a more thorough discussion of this limitation?

Q2: Besides SAR→RGB, the evaluation relies exclusively on corruption benchmarks with simple transformations that are trivially easy to learn for a transport map. It would be interesting to see results on another real cross-domain dataset or a more complex artificial cross-domain dataset where the transport map faces a genuinely challenging reconstruction problem. This would give a much clearer picture of how well the framework holds up in realistic domain transfer scenarios.

Q3: Can the authors explain why this conservativeness persists in low-mismatch regimes, and is there a way to make TCC-KS less conservative when the detected mismatch is small?

Q4: It is not clear from the paper in which settings the proposed correction mechanisms actually provide a meaningful practical benefit over simply applying standard split conformal prediction after transport. Could the authors clarify this?

I am willing to raise my score if the authors adequately address the concerns raised in this review, in particular Q2 and Q4.

**Limitations:**

yes

**Strengths And Weaknesses:**

## **Strengths**

**S1: Clearly defined and well-motivated problem setting.** The paper identifies a concrete gap in the conformal prediction research, where labeled calibration exists only in a source domain but predictions are needed in a different target domain. The problem is concisely formalized.

**S2:  Solid theoretical foundation.** The authors prove that without additional structural assumptions, no label-free procedure can achieve non-trivial distribution-free coverage guarantees in the worst case. This is a meaningful theoretical contribution and motivates why additional assumptions are unavoidable. Furthermore, the paper provides high-probability target coverage guarantees for TCC-KS that adapt to the observable mismatch certificate $\delta^+$, and an oracle-weighted validity statement for weighted-TCC. Together, these results give the approach a principal theoretical foundation that goes beyond purely empirical evaluation.

## **Weaknesses**

**W1: Strong paired data assumptions.** The method requires unlabeled paired observations where each pair corresponds to the same underlying instance. While the authors motivate this through sensor transitions and modality changes, such perfect pairing is rarely available in practice. The paper would benefit from a more critical discussion of this limitation and experiments with imperfect or approximate pairing would substantially strengthen the paper's claim.

**W2: Weak experimental evaluation.** The corrupted image benchmarks used in the evaluation are not representative of realistic domain transfer scenarios, since pairs are perfectly aligned by construction, and the transformations are simple analytical operations such as brightness shifts or additive Gaussian noise. This is also reflected in the results. $\delta^+$ is an order of magnitude smaller on the corruption benchmarks compared to the real cross-modal setting SAR->RGB, suggesting that the corruption benchmarks do not adequately stress-test the framework. The authors also acknowledge this in Section 5. However, the evaluation only includes one such realistic scenario. More realistic evaluations with complex or composed transformations would substantially strengthen the paper's claim.

**W3: Unnecessary conservativeness of TCC-KS.** Even in settings where $\delta^+$ is very small (0.01-0.03), TCC-KS systematically over-covers and produces substantially larger prediction sets than Transported CP. This suggests that the conservativeness is structural rather than truly driven by the detected mismatch.

**W4: Marginal benefit of the proposed corrections.** In the main results (Table 1), transported CP (the simplest baseline without any correction) performs comparably to weighted-TCC across all datasets and α levels and outperforms TCC-KS. The authors themselves note in Section 4.1 that "Transported CP and weighted-TCC are close to nominal". However, if transport alone already achieves near-nominal coverage, the practical benefit of the proposed correction technique becomes unclear. The stress test (Table 2) shows some separation, but even there the improvements are modest. This raises concerns about the overall contribution If the main insight reduces to learning a transport map and applying standard split conformal calibration, the technical novelty appears limited.

---

> ### Author Rebuttal · Authors · 2026-03-30
>
> We thank the reviewer for the careful and constructive review. Below, we respond in the same order as the reviewer’s weaknesses and key questions.
> ## Response to W1 / Q1
> We appreciate this question. In TCC, imperfect pairing affects the method primarily **through transport quality**: if the source-target correspondences used to learn $f$ are weaker or noisier, the learned map degrades, the transported distribution $\widetilde X_t=f(X_s)$ moves farther from the true target distribution, and the residual post-transport mismatch that TCC-KS and weighted-TCC must correct becomes larger.
>
> This is already reflected in our transport-quality study (Section 4.2 / Table 3): as transport improves, $\delta^+$ decreases and ESS increases, yielding tighter TCC-KS sets and more reliable weighted-TCC behavior. For example, on Tiny-ImageNet-C, $\delta^+$ drops from 0.1006 to 0.0148 while ESS rises from 48.4 to 91.8; on CIFAR-100-C, $\delta^+$ drops from 0.0521 to 0.0152 while ESS rises from 72.9 to 93.3. We will clarify this more explicitly in the revised discussion and limitations section.
>
> ## Response to W2 / Q2
> We contribute the results of a new experiment conducted on the **SUN RGB-D**  benchmark as a new real paired benchmark for 19-class indoor scene classification, using RGB images as source and depth images as target, with the same training and evaluation protocol as in SEN12MS. For space reasons, we report results for $\alpha=0.1$. For $\alpha=0.2$, we refer to **Response to Weakness 3 for Reviewer zdBJ**.
>
> | Method | Coverage |
> |---|---:|
> | Oracle CP | 0.902 (2.66) |
> | Transported CP | 0.875 (2.58) |
> | TCC-KS | 0.918 (3.62) |
> | weighted-TCC | 0.895 (2.79) |
> | WCP | 0.931 (4.95) |
>
> This benchmark remains realistic: RGB and depth are naturally paired modalities used together in real applications. In this setting, the same intended pattern remains visible: TCC-KS reaches slightly above-target coverage with a reasonable increase in set size, whereas WCP is substantially more conservative. At the same time, weighted-TCC remains close to Transported CP and recovers most of the remaining gap efficiently.
> ## Response to W3 / Q3
> We would like to highlight that TCC-KS is conservative because it (i) applies a one-sided correction through $\delta^+$ and (ii) includes finite-sample inflation to certify that correction with high probability. So the over-coverage is not a structural artifact unrelated to mismatch; it is the cost of a validity-oriented safeguard.
>
> The results already support this. In Table 4, the representative **mild-mismatch** case has $\delta^+=0.023$, for which Transported CP achieves 0.905 / 32.6 and TCC-KS 0.931 / 38.7, i.e. only 1.26× set-size inflation. Likewise, in Table 3 the best +KL / epoch-150 checkpoints yield very small mismatch on CIFAR-100-C and Tiny-ImageNet-C ($\delta^+=0.0152$ and 0.0148), where weighted-TCC is near nominal and efficient, while TCC-KS remains modestly conservative.
> ## Response to W4 / Q4
> The benefit of our correction layer is theoretically universal at the level of guarantees and most visible empirically when residual mismatch after transport is nontrivial.
>
> On the theoretical side, once calibration is transported across spaces, standard split CP no longer provides a target-domain guarantee. This is exactly why our Theorem 3.1 shows that, without additional structure, nontrivial label-free transfer is impossible in general, and why Theorem 3.2 is needed: it restores a target-domain validity statement for TCC-KS under explicit assumptions and an observable mismatch certificate. So even when Transported CP looks numerically close to nominal, it does **not** come with the same guarantee.
>
> On the empirical side, the practical value becomes clearest precisely in the regimes where transport is not sufficient on its own. This is what **Table 2** isolates. At $\alpha=0.1$, on CIFAR-100-C, Transported CP and weighted-TCC both under-cover at 0.88, whereas TCC-KS restores coverage to 0.91. Importantly, WCP also restores coverage, but with much larger sets (53.10), so TCC-KS achieves the correction at a clearly smaller efficiency cost.
>
> This is not isolated to one summary table. In Appendix F, where we characterize hard regimes in which even Assumption A2 is does no longer hold, we show that across the identified under-coverage settings, transport-only reaches 87.9% coverage, whereas TCC-KS recovers in 100% of the cases.
>
> So the practical benefit is regime-dependent but clear: weighted-TCC is the more efficiency-oriented correction when weighting remains stable, while TCC-KS is the validity-oriented correction when residual mismatch becomes hard. In addition, the new SUN RGB-D benchmark confirms the same behavior in a realistic cross-modal setting: at $\alpha=0.1$, Transported CP reaches 0.875, weighted-TCC 0.895, and TCC-KS 0.918.
>
> Once again, we thank the reviewer and hope the clarifications above address the concerns. If so, we would be grateful for a reconsideration of the score.

---

> > ### Author Rebuttal · Reviewer_6gcV · 2026-04-02
> >
> > I thank the authors for their thorough response. The new SUN RGB-D experiment directly addresses my main concern (Q2) and the clarification on Q4 strengthens the motivation of the approach. I raise my score from 3 to 4 accordingly.

---

> > > ### Author Response · Authors · 2026-04-05
> > >
> > > Thank you very much for your thoughtful follow-up and constructive feedback. We are very glad that our rebuttal addressed your main concerns, and we will incorporate the necessary modifications, additional experiment, and discussion based on your feedback.
> > >
> > > We just wanted to note, in case it is helpful, that the numeric score currently still appears as 3 on the review form. We would be grateful if the numeric score could be aligned with the written assessment. Thank you again for your time and careful reading.

---

### Official Review · Reviewer_GTgY · 2026-03-12

**Soundness:** 3
**Presentation:** 4
**Significance:** 2
**Originality:** 3
**Overall Recommendation:** 4
**Confidence:** 3

**Summary:**

The authors are addressing when the calibration data and test data are coming from separate modalities -- different resolutions, sensor results, etc. The main question is that with the help of unlabeled instances under both modalities how can we transfer the calibration from the first distribution to the second. They propose two approaches:

**Compliance With Llm Reviewing Policy:**

Affirmed.

**Final Justification:**

In light of the reviewer’s answers to my questions, I increase my score. However still I recommend the authors to elaborate more about the distribution of coverage over different methods, as it plays an important role in proving their point.

**Key Questions For Authors:**

I have doubts about assumption 2. First what happens if the domains are completely different; e.g. source is the video, and target is an image. Furthermore the authors are assuming that the largest distance in the true class is always less than the largest distance in the top class + a known offset. I can not see why in cases this can be true unless we allow the offset to be sufficiently large.

**Evaluation**
1. It is still surprising for me why in the empirical result, other CP methods are also doing well. For a alpha = 0.1, the coverage will shift randomly between 89~91 and this applies for all methods. Intuitively I was at least expecting for the conventional CP to break severely under this shift in the domain. Can you elaborate?

2. On the set size, specifically on datasets with larger set sizes, it can be possible that the distribution of set sizes over points can be better in method A compared to B, while B shows a smaller average set size. This can potentially happen when there are few large prediction sets in A that exceed in size drastically, meanwhile the rest of the datapoints are predicted with smaller sets. I ask the authors to report the distribution of set sizes over test points (maybe with a box plot, or histogram).

3. Can you elaborate how from table 1, or 2 one can understand that your method is improving upon baselines, and fixing an issue in the naive approach?

**Limitations:**

I think the paper is nice theoretically, but looking at the empirical results I am confused if the method is doing what I understood it should do.

**Strengths And Weaknesses:**

**Strength**

1. The problem is very interesting to investigate. I can see many applications with such usecase.

2. The approach TCC-KS is both interesting and intuitively nice. It is also mathematically rigorous. I think the paper in general targets and important problem for which the authors propose a nice and applicable framework, and solve the problem nicely.

**Weaknesses:**

1. Writing issues: in the introduction the authors used \tilde X without discussing what it refers to. I guess that is a target X generated by the mapping, but this is known after reading section 3. However the rest of the paper flows nicely.

2. The empirical evaluation does not fully show that the method is better than other approaches. I have discussed this in detail in the questions.

---

> ### Author Rebuttal · Authors · 2026-03-30
>
> We thank the reviewer for the thoughtful comments. Below, we respond in the same order as the concerns raised in the review.
>
> ## Question 1: on Assumption A2
> We appreciate the opportunity to clarify that A2 is a **distribution-level proxy condition** relating score-distribution mismatch to observable surrogate mismatch.
> For completely different domains, we do **not** claim that informative label-free transfer is always possible; our impossibility theorem makes exactly this point. In such regimes, A2 may fail or hold only with large slack, and TCC-KS is designed to respond conservatively rather than over-claim validity. Operationally, TCC-KS sets $\alpha^\star=\max(0,\alpha-\delta^+)$, so it stays close to transported split conformal when mismatch is small and becomes more conservative when mismatch grows. If $\delta^+\ge\alpha$, it enters a fallback regime ($\alpha^\star=0$) whose role is to avoid under-coverage rather than preserve efficiency, while maintaining the target-domain validity statement under the stated assumptions.
>
> This is also consistent with Appendix F, where we characterize A2 violations and failures. Across 328 configurations, the identified under-coverage cases all occur in moderate to hard regimes, where transport-only reaches 87.9% coverage versus 89.9% for oracle CP, while TCC-KS recovers 100% of the cases. This is the intended behavior: when the domains remain alignable, the mismatch certificate stays small; when they become too different, it grows and TCC-KS switches into a stronger guardrail mode.
>
> ## Eval Question 1
> The main reasons why the baselines are doing well in Table 1, is that Table 1 averages over all modes, including many mild settings (severities 1–2, and often part of 3) where transport already reduces residual mismatch enough that the baselines remain close to nominal on average.
>
> The clearer behavior appears once we isolate the harder residual-mismatch settings in **Table 2**: under strong corruptions with high severities 4–5, Transported CP and weighted-TCC under-cover, while TCC-KS restores nominal coverage. At $\alpha=0.1$, on CIFAR-100-C they drop to 0.88 and 0.88, whereas TCC-KS recovers to 0.91.
>
> The same pattern appears in the per-corruption/severity appendix tables. For example:
> - CIFAR-100-C, contrast, severity 5 (Appendix C / Table 7): Transported CP is 0.89, WCP-no-transport is 0.84, weighted-TCC is 0.90, while TCC-KS reaches 0.95.
>
> Appendix F confirms this systematically. Across the 328 audited configurations, the identified under-coverage cases are all concentrated in the harder regimes (early-to-mid training, motion blur / shot noise, severities 4–5). Across these 11 settings, transport-only reaches 87.9% coverage versus 89.9% for oracle CP, while TCC-KS recovers, with a 100% success rate.
>
> ## Eval Question 2
> To address this in the text-only rebuttal format, we report a numerical box-plot summary for set size on CIFAR-100-C, contrast, severity 3, $\alpha=0.1$, with $\delta^+=0.026$.
> | Method | Mean | Median | P90 | Max |
> |---|---:|---:|---:|---:|
> | Oracle CP | 30.64 | 31 | 50 | 71 |
> | weighted-TCC | 31.94 | 32 | 52 | 74 |
> | Transported CP | 32.62 | 33 | 53 | 76 |
> | TCC-KS | 38.73 | 40 | 58 | 80 |
>
> This shows that the larger mean set size of TCC-KS reflects a controlled conservative correction. At the same time, weighted-TCC remains very close to Transported CP across mean, median, and upper tail, consistent with its role as the more efficiency-oriented correction when weighting remains stable.
>
> ## Eval Question 3
> The intended reading is:
> - **Table** 1 is an average-case summary over many corruptions and severities. Because many of these settings are mild after transport, Transported CP and weighted-TCC often remain close to nominal on average. This shows that transport can already be effective, but it does **not** imply that the naive approach is valid in general.
> - **Table 2** is the key table for seeing the actual failure mode and the benefit of our method. It highlights the **harder residual-mismatch regime** (e.g., shot noise and motion blur at severities 4–5), where transport-only calibration is no longer reliable: Transported CP and weighted-TCC both under-cover at 0.88, whereas TCC-KS restores coverage to 0.91. WCP also restores coverage, but with substantially larger sets, so TCC-KS achieves the correction at a clearly smaller efficiency cost.
>
> This is also consistent with the appendix results: Appendix C shows concrete severe cases where transport-only under-covers (e.g., CIFAR-100-C contrast severity 5: 0.89 for Transported CP vs. 0.95 for TCC-KS), and Appendix F shows that across the 11 identified under-coverage settings, transport-only reaches 87.9% coverage while TCC-KS recovers in 100% of the cases.
>
> We thank the reviewer again for the feedback, and we hope our clarifications address the concerns. If the reviewer finds that these points have been satisfactorily resolved, we would be grateful for a reconsideration of the score.

---

> > ### Author Rebuttal · Reviewer_GTgY · 2026-04-03
> >
> > Thanks for the reply.
> >
> > - Regarding the question 1, I am not convinced that the other methods are partially failing. Still it is a question that why their failure is mild but I am convinced enough to accept the point.
> >
> > I strongly recommend the authors to add plots regarding the distribution of coverage for data in table 1, and 2. As you mentioned it is an aggregation of various modes, therefore seeing the variance can be very helpful.
> >
> > I increase my score.

---

> > > ### Author Response · Authors · 2026-04-05
> > >
> > > Thank you very much for the thoughtful follow-up and for increasing the score. We appreciate your comments and are glad that our response helped clarify the main point. We will ensure to incorporate the necessary modifications based on the feedback, including adding plots to better show the variance across the different modes.

---

### Decision · Program_Chairs · 2026-04-30

**Decision:**

Accept (regular)

**Comment:**

This paper tackles a highly practical and novel problem: transferring conformal prediction guarantees from a labeled source domain to an unlabeled target domain using paired observations (e.g., cross-sensor or cross-modal settings). To address this, the authors propose Transported Conformal Calibration (TCC), supported by two complementary correction mechanisms (TCC-KS and weighted-TCC).

Following the rebuttal and discussion phase, the reviewers reached a unanimous consensus to accept the paper. The recommendation for acceptance is driven by the following key strengths:

*   **Novel and Well-Motivated Setting:** The cross-space conformal calibration transfer problem is fresh and addresses a realistic bottleneck in deploying uncertainty quantification across different modalities or upgraded sensors.
*   **Rigorous Theoretical Foundation:** The theoretical development is highly praised. The impossibility theorem cleanly justifies the need for structural assumptions, and the finite-sample target coverage bounds tied to observable mismatch certificates provide a principled guarantee.
*   **Strong Rebuttal and Improved Evaluation:** Initially, several reviewers expressed concerns about the reliance on synthetic corruption benchmarks and questioned the practical benefit of the correction mechanisms over naive transport. The authors provided an excellent rebuttal, most notably by introducing a new, realistic cross-modal experiment on SUN RGB-D. This new benchmark effectively demonstrated the necessity and superiority of the proposed method in real-world scenarios.
*   **Practical Diagnostics:** The label-free diagnostics (such as the mismatch certificate for TCC-KS) provide actionable signals for practitioners to detect when calibration transfer might fail, adding significant practical value.

The submission is borderline. While the reviewers were all positive, there was not enough enthusiasm to recommend acceptance.